# Biomimetic NIR-II fluorescent proteins created from chemogenic protein-seeking dyes for multicolor deep-tissue bioimaging

Jiajun Xu[1,2,3], Ningning Zhu[1,2], Yijing Du[1,2], Tianyang Han[1,2], Xue Zheng[1,2], Jia Li[1] & Shoujun Zhu [1,2] ✉

Near-infrared-I/II fluorescent proteins (NIR-I/II FPs) are crucial for in vivo imaging, yet the current NIR-I/II FPs face challenges including scarcity, the requirement for chromophore maturation, and limited emission wavelengths (typically < 800 nm). Here, we utilize synthetic protein-seeking NIR-II dyes as chromophores, which covalently bind to tag proteins (e.g., human serum albumin, HSA) through a site-specific nucleophilic substitution reaction, thereby creating proof-of-concept biomimetic NIR-II FPs. This chemogenic protein-seeking strategy can be accomplished under gentle physiological conditions without catalysis. Proteomics analysis identifies specific binding site (Cys 477 on DIII). NIR-II FPs significantly enhance chromophore brightness and photostability, while improving biocompatibility, allowing for high-performance NIR-II lymphography and angiography. This strategy is universal and applicable in creating a wide range of spectrally separated NIR-I/II FPs for real-time visualization of multiple biological events. Overall, this straightforward biomimetic approach holds the potential to transform fluorescent protein-based bioimaging and enables in-situ albumin targeting to create NIR-I/II FPs for deep-tissue imaging in live organisms.

Fluorescence imaging has received extensive attention in diverse multidisciplinary fields as a cutting-edge imaging technology due to its high sensitivity, excellent spatial-temporal resolution, and strong specificity[1-7]. Although it has been successful in unraveling the intricacies of biological systems, intravital deep-tissue imaging remains a significant challenge due to the optical opacity of biological tissues[8,9]. Recent advancements in laser and detector manufacturing have allowed the successful extension of linear and nonlinear fluorescence imaging to the second near-infrared (NIR-II, > 900 nm) window, which is considered "tissue-transparent"[10-15]. This has effectively resolved the optical access disorder in deep tissue, creating an avenue for clinical diagnosis and prognosis[16,17]. However, achieving accurate visualization of biological events primarily depends on developing high-

performance fluorophores[18]. Organic small-molecule fluorophores have been favored candidates for preclinical and clinical practice owing to their low toxicity[19-22]. Unfortunately, currently available NIR-II small-molecule fluorophores often lack appropriate optical and pharmacokinetic properties for clinical applications[23]. Therefore, there is an urgent need to develop improved methodologies to expand the NIR-II dye library further, thereby facilitating the early clinical translation of NIR-II bioimaging techniques.

With the advent and rapid expansion of NIR fluorescent proteins (FPs), researchers have been inspired to construct more NIR fluorescent probes[24-26]. The internal chromophore of FPs was itself almost non-luminous. The unique β-canister structure of FPs formed through biological self-assembly effectively restricts chromophore rotation

[1]Joint Laboratory of Opto-Functional Theranostics in Medicine and Chemistry, First Hospital of Jilin University, Changchun 130021, P.R. China. [2]State Key Laboratory of Supramolecular Structure and Materials, Center for Supramolecular Chemical Biology, College of Chemistry, Jilin University, Changchun 130012, P.R. China. [3]Key Laboratory of Medicinal Chemistry and Molecular Diagnosis of the Ministry of Education, Key Laboratory of Chemical Biology of Hebei Province, College of Chemistry and Materials Science, Hebei University, Baoding 071002, P.R. China. ✉e-mail: sjzhu@jlu.edu.cn

and reduces non-radiative transition energy dissipation. This results in the chromophore exhibiting bright luminescence[27–29]. Several series of infrared fluorescent proteins (iRFPs) have been reported, including iRFP670, iRFP682, iRFP702, iRFP713, and iRFP720, which exhibit exceptionally high brightness compared to other reported NIR FPs[30–32]. Recently, the concept of NIR-II bioimaging via the off-peak tail emission of iRFPs has been successfully validated[33]. However, these fluorophores are typically genetically encoded and must be generated through slow and oxygen-dependent maturation (ranging from minutes to hours) for bioimaging[34,35]. Scaling up the production of NIR FPs for direct high-dose in vivo bioimaging for individuals is also challenging[36,37]. Furthermore, FPs with tunable NIR-II peak emissions are currently unavailable, which significantly hinders the true benefits of NIR-II bioimaging.

Recently, fluorescent chemogenetic reporters consisting of synthetic organic dyes anchored covalently or non-covalently to gene-encoded protein tags have opened up prospects for on-demand bioimaging and biosensing. These hybrid systems combine the targeted selectivity of genetic protein tags and the advantages of synthetic organic fluorophores, which can provide improved optical properties covering most of the visible spectra[38,39]. Herein, inspired by this principle and an urgent desire for long-wavelength FPs, we successfully design and construct a biomimetic NIR-II fluorescent protein system. This system utilizes chemogenic protein-seeking dyes (e.g., NIR-II CO-1080 and other NIR-I/II dyes) as the chromophores, which are covalently bound to the cysteine in the hydrophobic cavity of proteins (e.g., human serum albumin, HSA) through nucleophilic substitution reaction. The hydrophobic cavity acts as a microreactor (proteins with favorable pocket conformation/thiol groups and dyes with appropriate structures), promoting the nucleophilic substitution reaction to occur under very mild conditions without the need for any catalysts. The HSA outer shell not only significantly enhances the brightness and photostability of the chromophores through the tight clamping effect, but also effectively compensates for any biocompatibility defects. Our biomimetic NIR-II FPs enable noninvasive and accurate NIR-II visualization of physiological and pathological conditions of the vascular and lymphatic system via simple intravenous injection, greatly facilitating intraoperative imaging-guided surgery and postoperative noninvasive monitoring. Utilizing the unique 1064 nm optimal excitation of the NIR-II FPs, multicolor imaging of biological events is realized in combination with the 808 nm excited NIR-I FPs. This work provides a proof-of-concept for the biomimetic NIR-I/II FPs, effectively overcoming the existing limitations of NIR FPs and potentially realizing the true benefit of NIR-II bioimaging.

## Results

### Chemogenic protein-seeking dyes for the construction of biomimetic NIR-II FPs

We proposed a chemoselective protein-seeking strategy for constructing NIR-II FPs from a bionics perspective. We first synthesized the NIR-II protein-seeking dye (CO-1080, Supplementary Fig. 1), with maximal absorption and emission wavelengths of 1044 nm and 1079 nm, respectively. CO-1080 dye was then chosen as NIR-II chromophore to explore matching protein shells (Fig. 1a). Three proteins (β-Lactoglobulin (β-LG), bovine serum albumin (BSA), and human serum albumin (HSA)) were selected as protein shells due to their favorite pocket conformation along with site-specific thiol (-SH) groups. The pocket can serve as a microreactor for -SH covalently binding with CO-1080 under very mild conditions (e.g., mixing). In detail, CO-1080 entered the protein pocket through supramolecular interactions and underwent covalent binding via a nucleophilic substitution reaction between the -SH group in proteins and the Cl-C bond in CO-1080 (we named it "protein-seeking strategy"). This supramolecular-enhanced process results in the creation of superbright biomimetic NIR-II FPs with CO-1080 chromophores being strictly confined. Among the three proteins examined, HSA@CO-1080 FPs showed the best NIR-II luminescence ability, with fluorescence enhancement up to 22.13-fold (Fig. 1b–d and Supplementary Fig. 4).

To better understand the binding mechanism of different proteins with NIR-II chromophores, a glide program-based non-covalent molecular docking simulation was carried out. As shown in Fig. 1e–g, HSA had the most amino acids involved in binding, including Arg472, Lys475, Thr478, Ser480, and Asn483. Meanwhile, the docking score and binding energy between three proteins and CO-1080 chromophore were calculated by the MM/GBSA method (Fig. 1h, i). It was found that HSA and CO-1080 chromophore showed the best docking score and highest binding energy, explaining why HSA had the greatest enhancement in the luminescence ability of CO-1080 dye. HSA could effectively restrict the free twisting of CO-1080 and minimize energy dissipation caused by the non-radiative transition.

Next, SDS-PAGE gel electrophoresis indicated that both BSA@CO-1080 and HSA@CO-1080 FPs displayed bright fluorescent bands at the corresponding molecular weight, verifying comparable and high-efficiency covalent binding to the CO-1080 dye (Fig. 1j). Gel electrophoresis data suggested that while β-LG and CO-1080 exhibited a certain covalent binding ability, β-LG@CO-1080 FPs showed evident unreacted CO-1080 band at the lower molecular weight position. High-resolution mass spectrometry (HRMS) also confirmed the covalent binding between CO-1080 and the corresponding proteins, simultaneously highlighting the best covalent binding efficiency between HSA and CO-1080 (Supplementary Fig. 5). Collectively, HSA protein was the optimal candidate for CO-1080 tagging/targeting, resulting in a stable and bright NIR-II fluorescent protein.

### Reaction optimization and optical properties of the HSA@CO-1080 FPs

While it is generally true that all NIR-I/II FPs can be created under gentle physiological conditions given adequate reaction time, such as mixing at 37 °C, we have optimized the reaction conditions for constructing the HSA@CO-1080 FPs with maximum yield. This aims to achieve a nearly complete degree of reaction to facilitate scaling up production. NIR-II brightness and gel electrophoresis consistently verified optimal reaction conditions including a reaction temperature of 60 °C, reaction time of 2 h, and reaction ratio between HSA and CO-1080 of 1:1 (Supplementary Fig. 6). To better comply with in vivo imaging requirements, we also investigated the effect of reaction concentration on the luminescence properties of HSA@CO-1080 FPs. Supplementary Fig. 7 shows that at HSA and CO-1080 reaction concentrations exceeding 10 μM, the NIR-II fluorescence signal of HSA@CO-1080 FPs showed a nonlinear increase accompanied by a quenching effect and visible precipitation of unreacted CO-1080 dye at concentrations > 50 μM. Consequently, we adopted an ultrafiltration concentration strategy with low reaction concentration (10 μM) for high-concentration NIR-II FPs preparation (Supplementary Fig. 8a). The size and morphology of HSA@CO-1080 FPs were highly homogeneous, indicating that installation of CO-1080 chromophore into the HSA cavity did not affect the structure and morphology of the free HSA protein (Supplementary Fig. 8b–d).

Next, we investigated the optical properties of the optimized HSA@CO-1080 FPs (Fig. 1k), and HSA@CO-1080 FPs exhibited the most significant fluorescence intensity under 1064 nm excitation, which was approximately 2.97-fold of 980 nm excitation and around 20.36 times greater than that of 808 nm excitation. HSA@CO-1080 FPs remained luminescent even when imaged at > 1500 nm imaging window. Fluorescence spectra and a series of in vitro penetration experiments verified that the HSA@CO-1080 FPs exhibited remarkable NIR-II brightness and penetration depth at 1064 nm excitation (Fig. 1l and Supplementary Figs. 9, 10). In addition, a comparison of the fluorescence spectra of free CO-1080 and HSA@CO-1080 aqueous solutions strongly confirmed the effectiveness of the biomimetic FPs strategy,

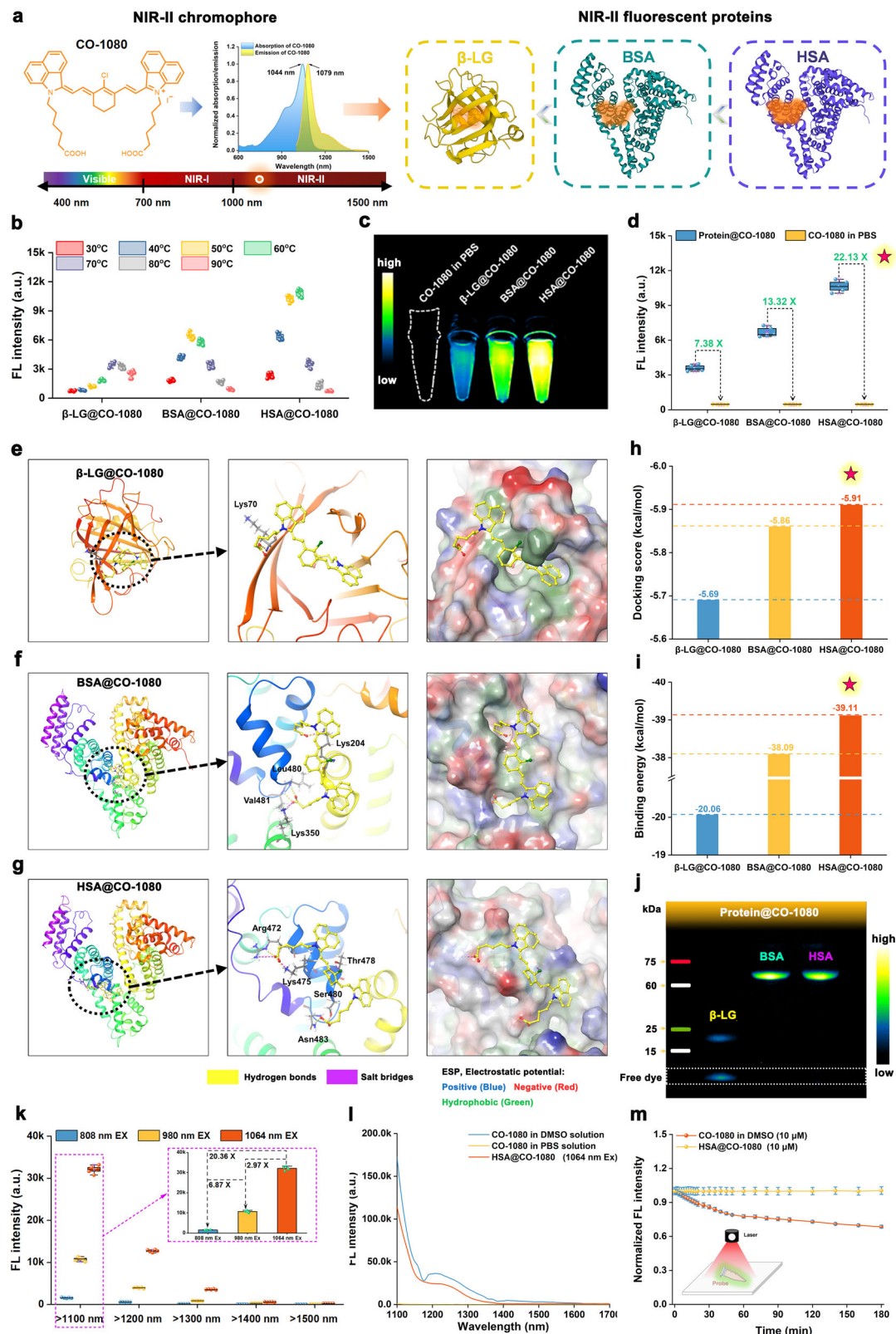

which greatly improved the NIR-II luminescence ability of CO-1080 chromophores, and even extended its imaging window to > 1500 nm. Excellent photostability is crucial for the potential clinical applications of contrast agents. Currently, clinically-approved indocyanine green (ICG) has been suffering from this issue. Installation of the protein-seeking CO-1080 in HSA@CO-1080 FPs effectively enhanced the photostability of the CO-1080 dye without any photobleaching

concerns (Fig. 1m and Supplementary Fig. 11). Overall, the satisfactory photostability of the HSA@CO-1080 FPs has great clinical translational potential.

### Chromophore optimization of biomimetic NIR-II FPs

To explore the best candidate of protein-seeking NIR-II dyes for constructing biomimetic FPs, we selected three 1080 analogues (Et-1080,

**Fig. 1 | Construction and characterization of biomimetic NIR-II fluorescent proteins. a** Schematic of NIR-II chromophore and NIR-II fluorescent proteins (FPs). **b** Brightness measurement of NIR-II FPs after CO-1080 installing into protein at various reaction temperatures (*n* = 5 independent samples per group). **c** NIR-II imaging and **d** fluorescence enhancement effect of the β-LG@CO-1080, BSA@CO-1080, and HSA@CO-1080 FPs (*n* = 5 independent samples per group). Theoretical simulation of CO-1080 binding to **e** β-LG, **f** BSA, and **g** HSA proteins by gliding docking mode. Comparison of **h** docking score and **i** binding energy between CO-1080 and three proteins, including β-LG, BSA, and HSA proteins. **j** Gel electrophoresis analysis of the β-LG@CO-1080, BSA@CO-1080, and HSA@CO-1080 FPs (*n* = 4 independent experiment). **k** Comparison of NIR-II brightness of HSA@CO-1080 FPs under different laser excitation (mean ± SD, *n* = 5 independent samples per group). **l** Fluorescence spectra of CO-1080 in DMSO solution, CO-1080 in PBS solution, and HSA@CO-1080 FPs under 1064 nm excitation. **m** Photostability of CO-1080 and HSA@CO-1080 FPs under 1064 nm excitation (mean ± SD, *n* = 3 independent samples per group). Protein structures were generated by the Protein Data Bank (PDB). Source data are provided as a Source Data file.

FD-1080, and CO-1080, Supplementary Figs. 2, 3) for further investigation. As depicted in Fig. 2a, b, the CO-1080 dye had brighter NIR-II brightness than Et-1080 and FD-1080 dyes. The highest occupied molecular orbital (HOMO) and the lowest unoccupied molecular orbital (LUMO) of the Et-1080, CO-1080, and FD-1080 dyes were calculated by density functional theory (Supplementary Fig. S12), and results verified that the CO-1080 dye possessed the smallest bandgap of 2.9497 eV, allowing it to exhibit a larger molecular emission redshift and brighter NIR-II fluorescence.

The HSA@CO-1080 FPs exhibited a more pronounced NIR-II brightness, approximately 3.51-fold that of HSA@Et-1080 and 2.72-fold that of HSA@FD-1080 (Fig. 2c, d). Moreover, the fluorescence enhancement effect of HSA on CO-1080 was as high as 21.90-fold, whereas the fluorescence enhancement effect on Et-1080 and FD-1080 was only 6.72 and 7.71, respectively (Fig. 2e). In addition, the solution color and UV-absorption/fluorescence spectra changed to varying degrees after HSA reaction with Et-1080, CO-1080, and FD-1080 dyes (Supplementary Fig. 13). This phenomenon indicated that the interaction between HSA and 1080 dyes was different, leading to the distinct luminescence properties of the synthesized NIR-II FPs. The UV-absorption and fluorescence spectra of Et-1080, CO-1080, and FD-1080 before and after HSA installation supported the effectiveness of the biomimetic FPs strategy (Supplementary Fig. 14). This approach effectively avoided the H or J aggregation of pure dye molecules in an aqueous solution and significantly improved their NIR-II emission ability (Supplementary Fig. 15). Among them, HSA@CO-1080 FPs showed the most significant improvement in absorbance and NIR-II fluorescence emission. The molar extinction coefficients and NIR-II quantum yields ($QY_S$) of HSA@CO-1080 (>1100 nm, $QY_S$ = 0.0921%) were also greatly higher than HSA@Et-1080 and HSA@FD-1080 (Supplementary Fig. 16, Supplementary Fig. 17, and Supplementary Table 1).

To gain a better understanding of the varying fluorescence enhancement effects of HSA on a series of 1080 chromophores, we first performed the Bio-Layer Interferometry (BLI) to evaluate the binding affinity between the HSA and 1080 chromophores. As shown in Supplementary Fig. 18 and Supplementary Table 2, the CO-1080 exhibited the most prominent binding affinity to HSA with $K_D$ of ~2.13 nM, which was far superior to Et-1080 (~14.5 nM) and FD-1080 (~82.6 nM). Meanwhile, we also conducted glide program-based non-covalent molecular docking simulation to analyze the binding behavior between HSA and Et-1080, CO-1080, and FD-1080 dyes. Results revealed that CO-1080 had the largest number of amino acid interactions, as well as the highest docking score and binding energy, surpassing the binding of Et-1080 and FD-1080 dyes (Fig. 2f–j). These simulation findings strongly implied that HSA exhibited stronger free-twisting restriction on the CO-1080 dyes compared to the Et-1080 and FD-1080 dyes. This led to a more effective reduction in energy dissipation caused by non-radiative transitions, thus resulting in the optimal NIR-II brightness of HSA@CO-1080 FPs. We further analyzed the covalent binding between HSA and 1080 dyes through SDS-PAGE gel electrophoresis (Fig. 2k) and high-resolution mass spectrometry (HRMS) (Fig. 2l and Supplementary Fig. 19). The fluorescence bands at the corresponding protein positions indicated that HSA had a more efficient covalent binding ability with CO-1080 dyes. HRMS results

confirmed that HSA could only covalently bind to one dye molecule, with HSA@CO-1080 demonstrating the highest covalent binding efficiency of 88.98%, while Et-1080 and FD-1080 dyes showed suboptimal binding efficiencies of 51.84% and 5.31% with HSA (Fig. 2l and Supplementary Fig. 19). These results collectively indicate that the CO-1080 dye was the most ideal chromophore candidate for constructing NIR-II biomimetic fluorescent proteins.

## Binding domain analysis between HSA and a series of 1080 chromophores

Following the determination of the optimal construction protocol for biomimetic NIR-II FPs, we further explored in detail the specific binding domains between HSA and 1080 chromophores. HSA is known to comprise three domains, namely domain I (DI), domain II (DII), and domain III (DIII) (Fig. 3a)[40,41]. Accordingly, recombinant DI, DII, and DIII proteins were reacted with a series of 1080 chromophores to explore the specific domains associated with different chromophores. As shown in Fig. 3b–d and Supplementary Figs. 20–22, the NIR-II fluorescence, SDS-PAGE gel electrophoresis, and HRMS consistently showed that, regardless of Et-1080, CO-1080, or FD-1080 dyes, all of them covalently interacted with DIII. Conversely, DI and DII showed negligible fluorescence enhancement and covalent binding ability. Further investigation revealed that CO-1080 and DIII exhibited highly efficient covalent binding capacity, with a binding fraction of up to 97.27% and almost no pure protein peaks detected (Fig. 3e). In contrast, Et-1080 and FD-1080 showed unsatisfactory covalent binding ability with DIII, with binding fractions of 34.34% and 15.12%, respectively. Thus, it can be concluded that the covalent binding site of 1080 dyes and HSA is located on DIII, which reaffirms that CO-1080 is the most effective luminescence center that matches with HSA.

Inspired by our previous reports that chlorine-containing cyanine dyes could covalently bind to the cysteine (Cys) residues of albumin through nucleophilic substitution[42,43], we selected L-cysteine hydrochloride molecules as Cys residue substitutes for HSA, and sought to validate covalent formation via displacement of chloride on 1080 dyes by thiol group on Cys residues under the base catalyst conditions. As described in Supplementary Fig. 23, liquid chromatography high-resolution mass spectrometry (LC-HRMS) characterization of the corresponding reaction products successfully demonstrated that the thiol group of L-cysteine could react with meso-chlorine on the unique carbon of cyclohexenyl ring via nucleophilic substitution. It should be emphasized that the success of the above displacement reaction heavily relied on the base catalyst and an overdose of L-cysteine molecules (10:1 reaction ratio) addition. Meanwhile, the Cys-blocked CO-1080 failed to covalent bind with HSA. To better simulate and understand the reaction in the protein solution, we also carried out the equivalent reaction of CO-1080 and Cys without other additives. The LC-HRMS results indicated there was almost no nucleophilic substitution between CO-1080 and Cys (Fig. 3g). The displacement of meso-chlorine between CO-1080 and HSA could be completed under mild PBS buffer conditions, and our protein-seeking approach explained that the DIII domain of the HSA acted as a microreactor, triggering a catalysis-free nucleophilic substitution by efficiently adjusting the CO-1080 dye conformation (Fig. 3h).

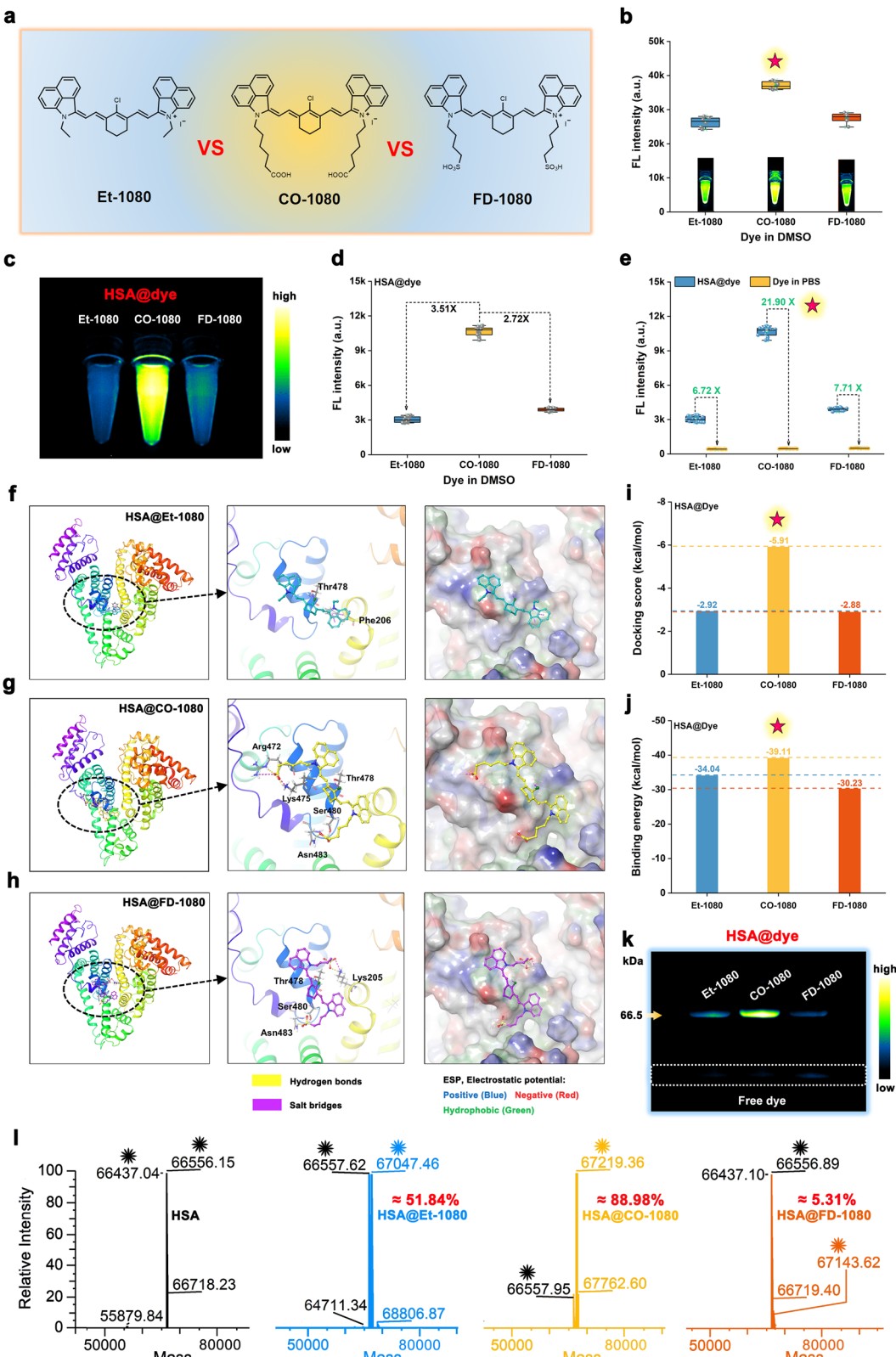

**Fig. 2 | Optimization of biomimetic NIR-II fluorescent proteins through chromophore selection. a** Chemical structure and **b** NIR-II brightness of the Et-1080, CO-1080, and FD-1080 chromophores (*n* = 5 independent samples per group). **c-d** NIR-II brightness and **e** fluorescence enhancement effect of HSA@Et-1080, HSA@CO-1080, and HSA@FD-1080 FPs (*n* = 15 independent samples per group). Theoretical simulation of HSA binding to **f** Et-1080, **g** CO-1080, and **h** FD-1080 chromophores by gliding docking mode. Comparison of **i** docking score and

**j** binding energy between HSA and three chromophores, including Et-1080, CO-1080, and FD-1080 chromophores. **k** Gel electrophoresis analysis of the HSA@Et-1080, HSA@CO-1080, and HSA@FD-1080 FPs (*n* = 4 independent experiment). **l** High-resolution mass spectrometry of the free HSA, HSA@Et-1080, HSA@CO-1080, and HSA@FD-1080. Protein structures were generated by the Protein Data Bank (PDB). Source data are provided as a Source Data file.

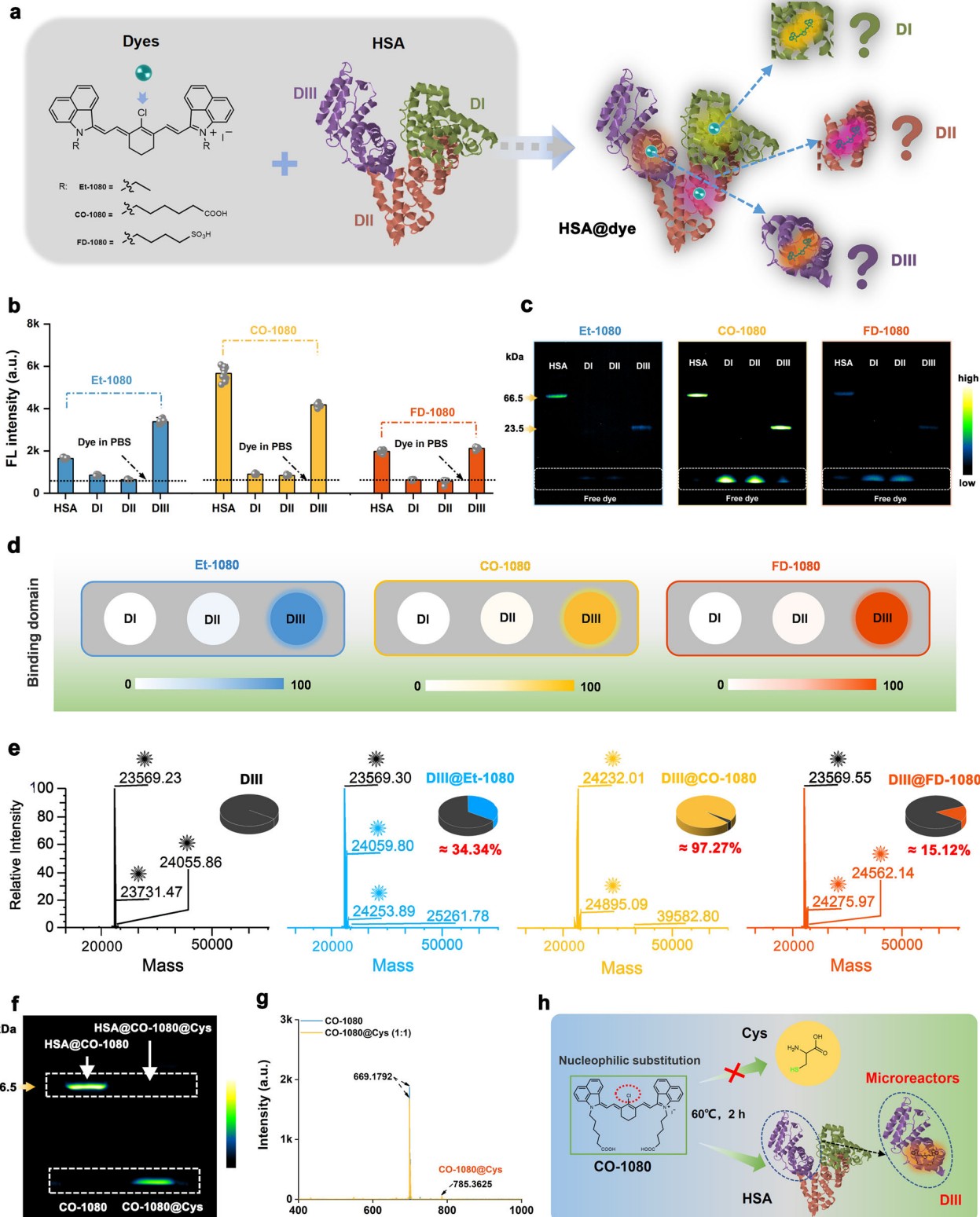

**Fig. 3 | Domain analysis of NIR-II chromophores binding to HSA. a** Schematic illustration of the binding domain between NIR-II chromophores and HSA. **b** Brightness (mean ± SD, $n = 10$ independent samples per group) and **c** gel electrophoresis analysis ($n = 4$ independent experiment) of the binding ability between Et-1080, CO-1080, FD-1080 chromophores and HSA, DI, DII, DIII. **d** Comparison of the covalent binding ability between three dyes and HSA, DI, DII, DIII. **e** High-resolution mass spectrometry of the DIII, DIII@Et-1080, DIII@CO-1080, and DIII@FD-1080. **f** Gel electrophoresis results of CO-1080 and CO-1080@Cys binding to HSA ($n = 4$ independent experiment). **g** Mass spectra of the CO-1080 and CO-1080@Cys obtained in a catalyst-free condition with a molar ratio of CO-1080 to Cys at 1:1(Cys: L-cysteine). **h** Reaction diagram showing CO-1080 with Cys and HSA (or DIII). Protein structures were generated by the Protein Data Bank (PDB). Source data are provided as a Source Data file.

## Specific covalent binding site analysis between HSA and CO-1080 chromophore

To identify specific covalent binding sites between CO-1080 chromophore and HSA as well as to understand the reaction mechanism, we performed an unbiased shotgun proteomics analysis (Fig. 4a). The NIR-II FPs (HSA@CO-1080) were first digested by chymotrypsin and trypsin, where specific cleavage sites were cut during the process (Supplementary Figs. 24, 25). The resulting peptides were then analyzed using ultra-performance liquid chromatography with tandem mass spectrometry (Nano LC-MS/MS). Afterward, the raw MS files were analyzed and searched against the target protein database using Byonic software, taking dye labeling as a variable modification ($C_{38}H_{46}N_2O_6S_2$, mass = 690.2797 m/z). The full mass (MS1) of all the expected peptides was calculated, and the residue composition of the peptide along with the exact binding residues was further inferred by analyzing the mass distribution of b and y series ions. The measurement accuracy for the fragments was < 0.02 Da (y ion series; y1–y8). As shown in Fig. 4b, c, and Supplementary Figs. 26, 27, the detailed proteomic analysis revealed four different peptide sequences with dye labeling, which corresponded to HSA protein residues Cys 477, Cys 361, Cys 487, and Cys 369. Notably, the sequence of K.CC[+ 661.307] TESLVNR showed the highest confidence in the downstream protein identification analysis compared with other detected sequences (Fig. 4d). In combination with the previous theoretical simulations and the validated covalent binding domain, it could be confidently predicted that the potential covalent binding site between HSA protein and CO-1080 was the Cys477 residue in the DIIIa sub-domain.

After identifying the specific covalent binding sites between HSA and CO-1080 chromophore, the next goal was to investigate their interaction and binding poses through covalent docking simulations. As shown in Fig. 4e–g and Supplementary Fig. 28, the most likely molecular docking conformations were successfully screened and the highest binding energy was observed between the HSA protein and CO-1080 chromophore. However, it was observed that the molecular covalent conformations were unstable under these conditions (Supplementary Movie 1). Therefore, we followed up with a molecular dynamics simulation (MD) analysis of the above-screened results using the Desmond module to determine the eventual stable HSA@CO-1080 conformation. Results indicated that the HSA and CO-1080 reached a steady state after 10 ns (Fig. 4e). After comprehensive consideration, MD simulation results at 30 ns were selected as the final covalent binding conformation (Fig. 4e–g and Supplementary Movie 1). Interestingly, it was discovered that the steady-state binding energy lay between the glide docking and covalent docking, and the molecular conformations under the MD simulation gradually approximated to be coplanar with the glide docking (Fig. 4g, h and Supplementary Fig. 29). Besides, the roles of different amino acids with CO-1080 chromophore during covalent conformational optimization were also analyzed in detail in Fig. 4i and Supplementary Fig. 30. Overall, it was hypothesized that the interaction between CO-1080 chromophore and HSA occurred in three successive stages: In stage I, the protein-seeking CO-1080 chromophore inserts into the hydrophobic pocket of HSA and binds to the pocket through supramolecular interactions. In stage II, a covalent bond is formed by the -SH group of Cys477 and the Cl-C group of the CO-1080 chromophore through nucleophilic substitution under the restricted microreactor of HSA. In stage III, the covalent conformation between CO-1080 chromophore and HSA undergoes fine-tuning to achieve the final steady-state bright-emitting NIR-II FPs. Stage III might occur simultaneously with stage II.

## Biosafety and pharmacokinetics of HSA@CO-1080 FPs

Considering their potential clinical application, it was necessary to evaluate the biosafety of HSA@CO-1080 FPs. We first evaluated the blood safety of CO-1080 chromophore and HSA@CO-1080 FPs (Supplementary Fig. 31a). Results confirmed even at high co-incubation concentration (200 µM), the CO-1080 chromophore and HSA@CO-1080 FPs had negligible hemolytic effects and were far superior to ICG. Further cell toxicity was assessed using fibroblast cell (L929) and breast cancer cell (4T1) types, with no apparent cytotoxicity observed even at a high co-incubation concentration of 200 µM for HSA@CO-1080 (Supplementary Fig. 31b, c). In contrast, the CO-1080 chromophore showed significant cytotoxicity at a co-incubation concentration of 30 µM, highlighting the rationalities and advantages of biomimetic FPs strategies to improve dye biocompatibility. Having established excellent biosafety in vitro, we then evaluated the in vivo biotoxicity of HSA@CO-1080 FPs by injecting them intravenously into mice. All biochemical parameters were within the normal value ranges, and no noticeable acute/chronic body injury was detected (Supplementary Fig. 31d–i). Rapid and controllable pharmacokinetics and minimal accumulation of contrast agents in major organs post-imaging are essential for clinical translation. As shown in Supplementary Fig. 32, both the CO-1080 chromophore and HSA@CO-1080 FPs had relatively low organ accumulation after tail vein injection and were able to be completely excreted within 7 days post-injection. Notably, the HSA@CO-1080 FPs exhibited a more prominent in vivo distribution and imaging capability in the short term compared with CO-1080. Collectively, the HSA@CO-1080 FPs demonstrated excellent biosafety and controlled pharmacokinetics, making them primed for in vivo imaging in NIR-II windows. Some aspects remain unclear, for example, the rapid decrease of in vivo brightness of HSA@CO-1080 at the initial time points needs further detailed investigation.

## NIR-II lymphography and angiography of HSA@CO-1080 FPs

Reliable visualization of lymphatic/vascular system visualization is crucial for assessing functionality and monitoring regenerative capacity in biomedical applications (Fig. 5a). Currently, although ICG lymphography has been used in clinical imaging, its imaging window remained in the NIR-I region or limited NIR-II window, resulting in limited penetration depth and resolution, which severely restricts the full potential of NIR fluorescence imaging technology[44,45]. To address these limitations, we investigated the ability of HSA@CO-1080 FPs to perform NIR-II lymphography. As shown in Supplementary Fig. 33a, HSA@CO-1080 FPs showed superior lymph node localization and lymphatic vessel delineation under 1064 nm excitation, outcompeting free CO-1080 chromophores. Notably, the HSA@CO-1080 FPs maintained excellent lymphatic imaging capability at > 1500 nm imaging window, thereby improving the imaging quality for the lymphatic system (Supplementary Fig. 33b). HSA@CO-1080 FPs also demonstrated significantly superior NIR-II lymphatic imaging capability and longer time window compared to the clinically ICG (Supplementary Fig. 34). NIR-II FPs were then successfully applied for image-guided precise dissection of lymph nodes and high-quality visualization of lymphedema diseases (Fig. 5b, c). Taken together, the HSA@CO-1080 FPs are prime candidates for the next-generation NIR-II lymphography, addressing the deficiency/limitations of clinical ICG lymphography.

Blood vessels are essential for delivering nutrients and maintaining the normal function of organs throughout the body. High-quality angiography can assist surgeons in more efficiently identifying vascular lesions in various organs and evaluating postoperative blood flow recovery[46,47]. Therefore, we conducted a series of preclinical angiography studies on HSA@CO-1080 FPs. As shown in Fig. 5d–f and Supplementary Fig. 35, HSA@CO-1080 FPs demonstrated superior systemic and local (leg) vascular NIR-II imaging capabilities compared to the CO-1080 chromophores. Shifting the imaging window to longer wavelengths further improved the imaging quality, particularly in > 1400 nm and > 1500 nm imaging windows, with almost no skin/background signal interference. In addition, 1064 nm excitation achieved a more prominent tissue penetration depth and spatial resolution, further highlighting the benefits of HSA@CO-1080 FPs over CO-1080 dye (Supplementary Fig. 36). During imaging, we observed that veins and

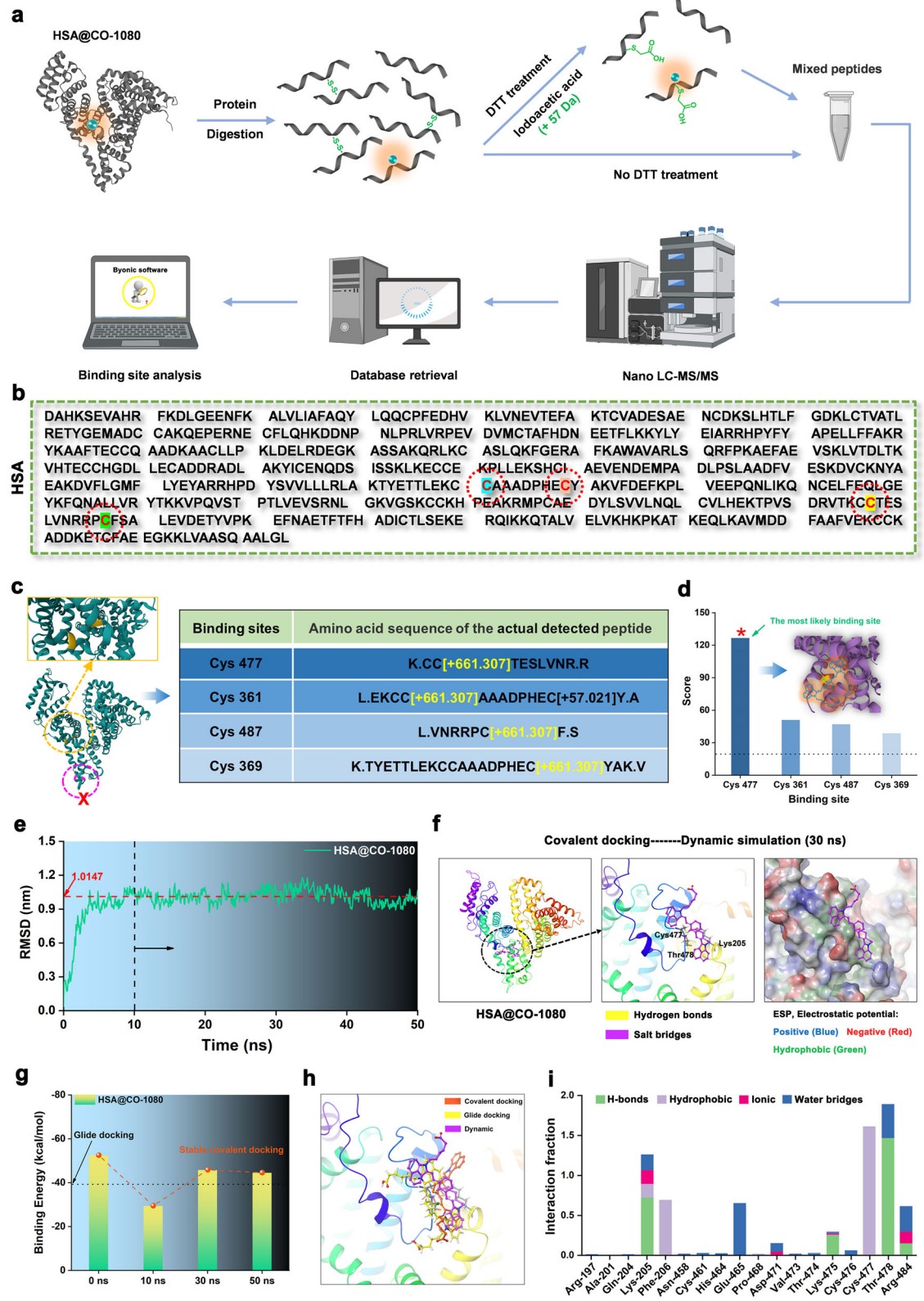

**Fig. 4 | Analysis of specific binding sites of the CO-1080 chromophore to HSA.**
**a** Proteomics analysis flowchart of HSA@CO-1080 FPs (FPs: fluorescent proteins, DTT: dithiothreitol). **b** Complete amino acid sequence of HSA. **c** Peptide information and specific binding sites containing CO-1080 chromophore obtained by proteomic analysis. **d** Score of all binding sites containing CO-1080 chromophore. **e** Dynamics simulation analysis of HSA@CO-1080 covalent docking. **f** Conformations of covalently docked HSA@CO-1080 at steady state (30 ns)

obtained by kinetic simulation. **g** Binding energy of CO-1080 in HSA at different times during the dynamic simulation. **h** Conformational comparison of CO-1080 in HSA under glide docking, covalent docking, and dynamic simulation.
**i** Contribution fraction of amino acid at different sites during the kinetic simulation and their interaction with CO-1080. All scale bar lengths represent 1 cm. Some schematic diagrams were designed using BioRender software. Source data are provided as a Source Data file.

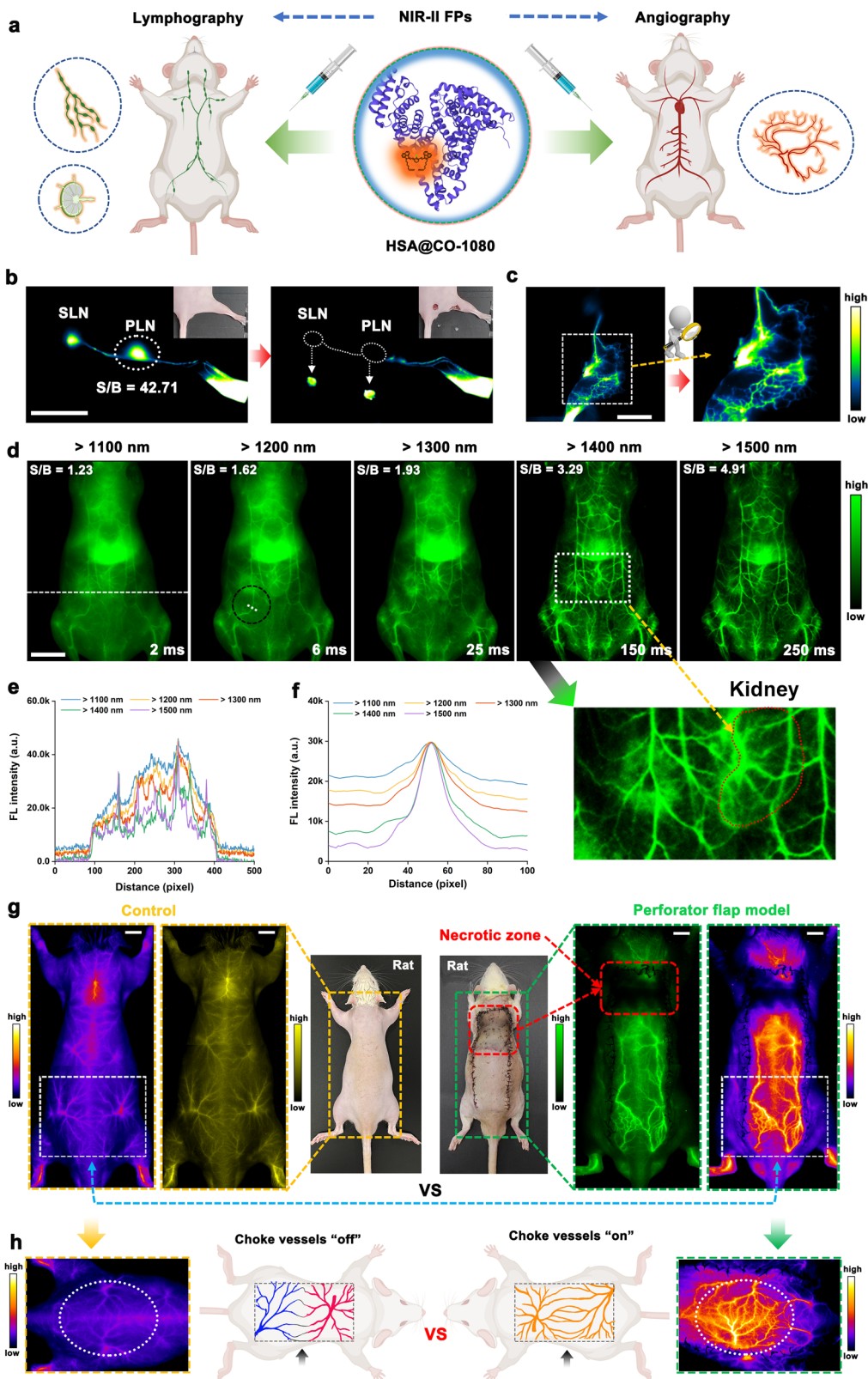

arteries could be clearly traced and distinguished following intravenous injection of the HSA@CO-1080 FPs (Supplementary Fig. 37a–c). Quantitative fluorescence signals were able to observe the flow rate and breathing frequency in the artery (Supplementary Fig. 37d, e). HSA@CO-1080 FPs were also successfully applied to achieve high-precision vascular imaging in relatively large mammals, including rats and rabbits (Supplementary Fig. 38). Besides, the blood half-life and

time window for angiography of HSA@CO-1080 FPs were described in Supplementary Figs. 39, 40. The above results consistently indicated that the HSA@CO-1080 FPs possessed rapid blood clearance and the fluorescence of the corresponding vessels could be ignored approximately 10 min after intravenous injection. It is important to note that estimating the overall excretion of HSA@CO-1080 FPs is challenging due to potential enzymatic degradation and/or dissociation between

**Fig. 5 | High-performance NIR-II lymphography and angiography using HSA@CO-1080 FPs. a** Schematic of the HSA@CO-1080 FPs for NIR-II lymphography and angiography. **b** HSA@CO-1080 FPs-guided lymph node imaging and NIR-II-guided surgical excision (PLN: popliteal lymph node, SLN: sacral lymph node, S: signal, B: background, > 1200 nm collection, 100 ms, *n* = 3 independent mice). **c** NIR-II lymphedema imaging of HSA@CO-1080 FPs (> 1200 nm, 100 ms, *n* = 3 independent mice). **d** NIR-II whole-body vessel imaging of HSA@CO-1080 FPs under different sub-NIR-II windows (*n* = 3 independent mice). **e, f** Cross-sectional fluorescence signals profile of the NIR-II whole-body vessels imaging. **g** Comparison

of the NIR-II angiography between normal rats and perforator flap model rats using HSA@CO-1080 FPs (*n* = 3 independent rats). **h** Comparison of the NIR-II choke vascular area imaging in rats before and after perforator flap modeling (> 1200 nm collection, 200 ms). The dosage of HSA@CO-1080 FPs was 600 μM (200 μL for mice and 1 mL for rats) for all imaging. For NIR-II lymphography, the HSA@CO-1080 FPs were injected into mice by intramuscular injection. For NIR-II angiography, the HSA@CO-1080 FPs were injected into mice via tail intravenous injection. All scale bar lengths represent 1 cm. Some schematic diagrams were designed using BioRender software. Source data are provided as a Source Data file.

the dye and albumin in vivo. Supplementary Fig. 41 displayed the investigation of the repeated angiography ability of HSA@CO-1080 FPs. As expected, the HSA@CO-1080 FPs could achieve multiple/long-term monitoring of blood vessels without any significant interference from skin signals, which also fully confirmed the ability of HSA@CO-1080 FPs for long-term detection/tracing of vascular-related diseases.

Due to the diversity of vascular dissection and complexity of perforator flap design, postoperative under perfusion can often result in local necrosis caused by ischemia and hypoxia, making flap necrosis the most common and serious complication of skin flap surgery[48–50]. Unfortunately, there are currently no appropriate imaging techniques for accurate and noninvasive monitoring of flap functionality. Our HSA@CO-1080 FPs successfully provided a high-contrast real-time monitoring of a modified perforator flap model postoperatively to simulate clinical flap transplantation (Supplementary Fig. 42). Unlike normal rats, the HSA@CO-1080 FPs enabled clear visualization of the postoperative recovery of the flap model and accurately localized the ischemic necrotic area (Fig. 5g). By intravenously injecting HSA@CO-1080 FPs and carefully observing the blood route of the flap area, it was found that the two reserved inferior gluteal artery (IGA) perforator vessels lit up first and then gradually spread upward (Supplementary Fig. 43). Compared with pre-modeling imaging results, the flap model group observed the opening of choke vessel and revascularization (Fig. 5h and Supplementary Fig. 44). In particular, the HSA@CO-1080 FPs successfully visualized the progression of the entire flap model due to the low background signals and fast excretion (capable of repeated observations). NIR-II imaging confirmed that the progression could be roughly divided into three stages: on postoperative day 2, the distal flap appeared significant swelling, turned dark purple, and began to develop ischemic necrosis. On postoperative day 4, the necrotic area of the distal flap was aggravated, with a dark brown color. On postoperative day 7, the distal flap developed a black scab with dry necrosis and a clear demarcation with the proximal side (Supplementary Fig. 45). Collectively, HSA@CO-1080 FPs possess excellent NIR-II angiography capability, which may promote the early clinical transformation of NIR-II fluorescence imaging technology and provide a noninvasive and effective method of monitoring flap functionality.

## Multicolor in-vivo imaging using a wide range of biomimetic NIR-I/II FPs

Due to the complexity and dynamic characteristics of organisms, static single-source signal acquisition cannot provide a complete picture of biological physiological, and pathological changes. Real-time multi-channel NIR-I/II imaging systems can record multiple events simultaneously, which could gain insight into the mechanism of disease occurrence, improve the imaging efficiency, and present high in vivo and clinical applicability[51,52]. Integrated fluorescence imaging for multiple biological events relies on non-crosstalk fluorescence probes. To validate the feasibility of the NIR-I/II FPs for multicolor bioimaging, we also selected NIR-I protein-seeking dyes, and successfully constructed biomimetic NIR-I FPs (HSA@IR-808 and β-LG@IR-780) and combined them with NIR-II FPs (HSA@CO-1080 and DIII@CO-1080) (Fig. 6a and Supplementary Fig. 46). The synthesized NIR-I and NIR-II FPs achieved non-overlapping emissions under 808 nm and 1064 nm excitation, respectively, providing their excellent two-color imaging potential

(Fig. 6b–d and Supplementary Fig. 47). The UV-absorption and fluorescence data under different laser excitations further confirmed the feasibility of using NIR-I and NIR-II FPs for two-color bioimaging (Supplementary Fig. 48). Subsequently, we conducted a series of in vivo multichannel bioimaging experiments using the NIR-I and NIR-II FPs. Firstly, an integrated multi-channel NIR-II lymphatic imaging was successfully achieved using the HSA@IR-808 FPs and the HSA@CO-1080 FPs (Fig. 6e–g and Supplementary Fig. 49). These NIR-I/II FPs successfully achieved the precise co-localization of tumor-infiltrating sentinel lymph nodes, thus guiding efficient surgical excision (avoiding unnecessary excision) intraoperatively (Fig. 6f, g). The integrated NIR-II bioimaging of vessels and lymphatic systems without crosstalk was also achieved using HSA@IR-808 FPs and the HSA@CO-1080 FPs (Fig. 6h and Supplementary Fig. 50), which is important for precise imaging-guided surgery. The combination of HSA@CO-1080 FPs (hepatobiliary excretion) and β-LG@IR-780 FPs (renal excretion) enabled simultaneous visualization of two pharmacokinetic pathways without crosstalk issue (Fig. 6i). Taken together, the NIR-I/II fluorescent proteins constructed based on biomimetic strategy possess excellent potential for multicolor bioimaging applications, promising to achieve integrated real-time imaging of multiple biological events and to better assist clinicians to handle the complex diseases.

Given the excellent two-color imaging properties of biomimetic fluorescent proteins, the next aim was to expand the imaging capabilities to include additional colors. For this purpose, we selected HSA@Cy5, β-LG@IR-780 FPs, RENPs (NaYbF$_4$:Ce, Er@NaYF$_4$:Gd, Yb@PAA), and HSA@CO-1080 FPs as candidates for multicolor bioimaging. We chose the RENPs because we lacked a sufficient number of spectrally distinct chromophores, which necessitates the synthesis of more dyes in future work. In-vitro imaging and UV-absorption/fluorescence data unanimously confirmed that the four selected probes were qualified for four-color imaging and could achieve bright luminescence with minimal crosstalk interference under their most suitable excitation/emission channels (Supplementary Fig. 51). We further verified the multicolor in vivo imaging ability of these probes in Fig. 6j. The experiment involved intravenous injection of β-LG@IR-780 FPs, followed by intragastric injection of RENPs and intraperitoneal injection of HSA@Cy5, and finally intravenous injection of HSA@CO-1080 FPs. The relative fluorescence signal contribution of the four channels (660 nm Ex, 808 nm Ex, 980 nm Ex, and 1064 nm Ex) helped determine the biological distribution of each probe, consistent with the images visualized in each corresponding single channel. Overall, the incorporation of NIR-I/II FPs has made NIR four-color in vivo real-time bioimaging a reality, promising to provide better insights into the mysteries of organisms and the mechanisms of diseases.

## Discussion

The discovery of the first fluorescent protein (avGFP) in the jellyfish Aequorea Victoria in 1994 marked the beginning of fluorescent proteins (FPs) as biomarkers for imaging Caenorhabditis elegans and Escherichia coli[53,54]. Building upon the structure and luminous mechanism of GFP, researchers have successfully extended the FPs spectra to orange, red, and near-infrared (NIR) regions (but still < 800 nm)[55–57]. Among these, NIR FPs have demonstrated superior

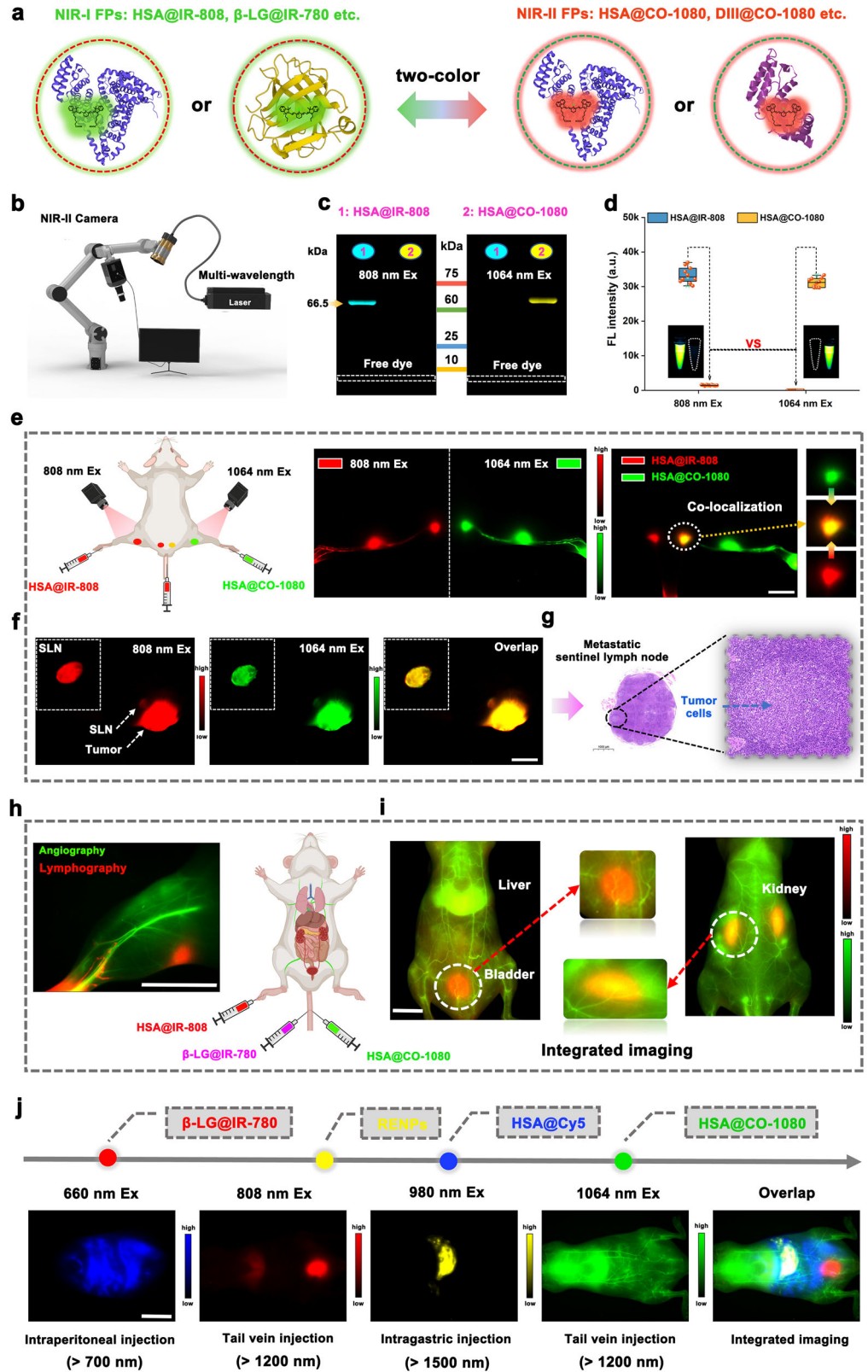

bioimaging capabilities compared to visible FPs. These NIR FPs were typically genetically encoded and must be genetically transfected into living cells and animals for biological imaging, and were generally accompanied by long maturation times, oxygen-dependent luminescence, or uncontrollable transfection efficiency[34,35,39,58]. Moreover, their imaging window remains in the NIR-I region, severely limiting the depth and quality of imaging. Recently, Qian et al. successfully validated the concept of NIR-II bioimaging using the emission tail of individual infrared fluorescent proteins (iRFPs), which was also the first application of NIR FPs in the NIR-II imaging window[33]. However, these gene-edited FPs cannot be scaled up under limited lab or industrial conditions, and were rarely reported for direct in vivo bioimaging via exogenous injection[36,37]. Additionally, the FPs with NIR-II peak emission remain absent so far.

**Fig. 6 | Biomimetic NIR-I/II FPs for multicolor in vivo imaging. a** Schematic of the biomimetic NIR-I FPs and NIR-II FPs constructed via biomimetic strategy. **b** Schematic of the NIR-I/II multi-wavelength excitation imaging device. **c)** Gel electrophoresis analysis of the HSA@IR-808 (NIR-I FPs) and HSA@CO-1080 (NIR-II FPs) ($n = 4$ independent experiment). **d** Fluorescence intensities of the HSA@IR-808 FPs and HSA@CO-1080 FPs under 808 nm and 1064 nm excitation, respectively ($n = 10$ independent samples per group). **e** Dual-color lymph node imaging and colocalization using HSA@IR-808 FPs (intradermal footpad or tail injection) and HSA@CO-1080 FPs (intradermal footpad injection) ($n = 3$ independent mice). **f** Co-localization imaging of tumor-associated sentinel lymph nodes using HSA@IR-808 FPs and HSA@CO-1080 FPs ($n = 3$ independent mice). **g** Pathological images of the tumor-draining sentinel lymph node ($n = 3$ independent mice). **h** Dual-color imaging of lymphatic system and blood vessels using HSA@IR-808 and HSA@CO-1080 FPs ($n = 3$ independent mice). **i** Dual-color metabolic behavior imaging using the β-LG@IR-780 FPs and HSA@CO-1080 FPs. **j** Multicolor in-vivo imaging using HSA@Cy5, β-LG@IR-780, RENPs (NaYbF$_4$:Ce, Er@NaYF$_4$:Gd, Yb@PAA), and HSA@CO-1080 FPs under different excitation wavelengths (660 nm, 808 nm, 980 nm, and 1064 nm) and collection wavelengths ($n = 3$ independent mice). Note: the injection concentration of the RENPs probe was 250 mg/ml ($> 1500$ nm collection), the injection concentration of the HSA@Cy5 probe was 200 μM ($> 700$ nm collection), and the injection concentration of the β-LG@IR-780, HSA@IR-808, and HSA@CO-1080 probe was 600 μM ($> 1200$ nm collections). All scale bar lengths represent 1 cm. Some schematic diagrams were designed using BioRender software. Source data are provided as a Source Data file.

In this work, inspired by the structure and luminous mechanism of conventional FPs, we designed and proposed a biomimetic fluorescent protein system with NIR-II peak emission, which combined the advantages of organic fluorophores with those of protein tags. We rationalized chemogenic protein-seeking NIR-II dye (CO-1080) as the chromophore, which can automatically covalently tag to the structure-appropriate proteins (e.g., HSA) via a nucleophilic substitution reaction to form the NIR-II FPs (HSA@CO-1080). Unlike traditional gene-coding construction strategies, HSA and NIR-II chromophores require only a simple mixed heating process to complete the construction of FPs. We conducted a series of in vitro experiments that demonstrated the effectiveness of this protein-seeking strategy in significantly improving the brightness, photostability, and biocompatibility of the chromophores. Therefore, this approach enables high-performance NIR-II lymphography (e.g., diagnosing lymphedema diseases) and angiography (e.g., monitoring flap functionality).

While the encapsulation of NIR-I/II fluorophores with proteins, whether exogenous or endogenous, has been proven as an effective approach to enhance their brightness and pharmacokinetics, the molecular mechanism underlying the interaction between proteins and dye molecules has been poorly understood and typically referred to as "complexes"[59–61]. We systematically investigated the interaction mechanism between proteins and NIR-II chromophores using docking modeling, domain-binding mass spectrometry, and proteomic analysis, ultimately determining the specific domains and binding site between HSA and CO-1080 (DIII, Cys 477). Comprehensive multi-angle research results suggested that the interaction between protein-seeking CO-1080 chromophore and HSA occurs in three successive stages, including supramolecular inserting of CO-1080, covalent binding, and the formation of optimal conformation between CO-1080 and HSA. Notably, our study represented the comprehensive exploration of the interaction between proteins and NIR-II cyanine dyes in this manner.

Considering the complexity and dynamic nature of organisms, single-color imaging is insufficient to fully reflect the physiological and pathological changes. Real-time multi-color NIR-II imaging systems could solve this issue through recording multiple events simultaneously, which not only could provide valuable insights into the workings of organisms and the mechanisms behind diseases, but also present great prospects in clinical practice for noninvasive diagnostics, image-guided surgery, etc[51,52,62,63]. In a word, the implementation of multi-color imaging could effectively improve bioimaging performance and imaging efficiency, with high in vivo and clinical applicability. However, due to the intrinsically broad linewidths and/or significant spectral overlap, only a handful of fluorescent probes with distinguishable features in the NIR-II imaging region have been reported[64]. Therefore, additional optical materials are needed to overcome the spectral limitation and break the "multiplexing ceiling". In line with this, we utilized biomimetic fluorescent proteins to construct several NIR-I FPs in synergy with NIR-II FPs, which enabled us to successfully perform integrated two-color NIR-II imaging of various biological events without interference, such as lymph node colocalization, lymphatic-vascular visualization, and pharmacokinetics assessment. In addition, by combining the synthesized NIR-I/II FPs with previously reported probes, we successfully developed a four-color imaging system without any crosstalk, effectively realizing integrated visualization of even more complicated biological events and getting a rainbow of color from a single protein tag.

In summary, we have demonstrated an alternative to conventional gene-coding techniques for developing NIR fluorescent proteins. Our biomimetic approach is both simple and effective, filling an important gap in the current limitations of NIR-II FPs to open up opportunities for fluorescent protein-based bioimaging. By screening a large number of protein-seeking NIR-I/II cyanine dyes with different optimal excitation/ emission wavelengths as chromophores to construct fluorescent proteins, we successfully achieved a multichannel imaging system, thus providing the possibility for integrated monitoring of various biological events. Ongoing development of cyanine dyes with more emission bands without crosstalk and longer emission wavelengths (e.g., > 1400 nm) will further enhance the enormous potential of our biomimetic fluorescent protein strategy. More importantly, this biomimetic protein strategy would also provide a fundamental principle for designing more dyes for in-situ protein targeting purposes.

## Methods

### Ethics statement

All animal experiments were conducted according to the protocols approved by the Animal Ethical Committee of The First Hospital of Jilin University (20210642).

### Materials

Human serum albumin (HSA, ≥ 98%), Bovine serum albumin (BSA, ≥ 98%), β-Lactoglobulin from bovine blood (β-LG, ≥ 90%), L-Cysteine (Cys, ≥ 98%), IR-780 (≥98%), and anhydrous dimethyl sulfoxide (DMSO, ≥ 99%) were purchased from Sigma-Aldrich. ICG (modified) for human injection was purchased from Dandong Yichuang Pharmaceutical Co. Ltd. IR-808 (≥ 95%) was purchased from Beijing Zhongying Biotechnology Co., Ltd. Cyanine5 NHS ester (Cy5, ≥ 99%) was purchased from Aladdin (Shanghai, China). Recombinant human albumin domain I (DI, ≥ 95%), recombinant human albumin domain II (DII, ≥ 95%), and recombinant human albumin domain III (DIII, ≥ 95%) were purchased from Albumin Therapeutics LLC. Benzo[cd]indol-2(1H)-one, phosphorus pentasulfide, pyridine, sodium hydroxide (NaOH), methyl iodide, acetone, methanol, 2,2-dimethyl-1,3-dioxane-4,6-dione, ethanol, triethylamine, ethyl iodide, N,N-Dimethylformamide (DMF), potassium carbonate, concentrated hydrochloric acid (HCl), potassium iodide, acetic acid, acetic anhydride, 1-iodopropane, 1,4-butylenesulfone, potassium hydroxide, N-methylpyrrolidone(NMP), 2,6-di-tert-butyl-4-methyl pyridine (DTBMP), toluene, 1-butanol, and ethyl 6-bromohexanoate were purchased from Energy Chemical.

### Instrument characterization

The UV-absorption spectra of different probes were measured using a LAMBDA 1050+ spectrophotometer. The NIR-II fluorescence spectra of

different probes were measured by Edinburgh instrument FLS920 fluorescence spectrophotometer. Sodium dodecyl sulfate-polyacrylamide gel electrophoresis (SDS-PAGE) was performed on different probes using an American BIO-RAD electrophoresis system. The covalent binding behavior of protein to different dyes was characterized by Orbitrap Eclipse high-resolution mass spectrometer. The binding behavior of L-Cysteine to different dyes was characterized by a high-resolution mass spectrometer (Agilent1290). The binding affinity between the HSA and 1080 chromophores was acquired through the ForteBio Octet Red96e molecular interaction analyzer (Biolayer interferometry technology, BLI). The particle sizes of different compounds were analyzed using a Malvern Zetasizer Nano ZS size analyzer. Transmission electron microscope (TEM) images were acquired through DTM-961002 of JEOL Ltd. The $^1$H-NMR spectra of all dyes were obtained on Bruker AVANCE III 400 MHz NMR spectrometers.

### Synthesis of CO-1080, Et-1080, and FD-1080 dyes
Firstly, 1,8-Naphthaolactam was alkylated under the protection of Meldrum's acid to obtain intermediates according to the previous literature[65–68]. In this process, two identical branched chains, including alkyl, carboxylic or sulfonic groups, were introduced to the indole ring of the intermediates. Subsequently, the alkylated compounds were deprotected to obtain the methylated intermediates. Finally, the modified intermediates were processed through the Knoevenagel reaction to obtain the corresponding target products (Et-1080, CO-1080, and FD-1080).

### LC-HRMS of CO-1080, Et-1080, and FD-1080 dyes
The mass spectrometry was operated under the specific conditions: ESI$^+$ spray voltage, 4.5 kV, or ESI-spray voltage, −3.5 kV; nebulizer gas, 1.5 L/min; drying gas, 100 kPa; heat block temperature, 200 °C; CDL temperature, 200 °C; IT Area Vacuum, $1.0 \times 10^{-2}$ Pa; TOF Area Vacuum, $5 \times 10^{-4}$ Pa. The ion accumulation time was set to 10 ms, and the detector voltage was fixed at 1.6 kV. The mass number calibration (ion trap and TOF analyzer) was completed using a solution of trifluoroacetic acid (TFA) and sodium hydrate. Data acquisition and analysis were performed using the LC-MS Solution version 3.0 software.

### Synthesis of NIR fluorescent proteins
All fluorescent proteins were synthesized using a similar protocol, with the HSA@CO-1080 probe being taken as an example here. First, 2 mM CO-1080 and 10 μM HSA were prepared in anhydrous dimethyl sulfoxide (DMSO) and PBS solutions, respectively. Next, 5 μL of 2 mM CO-1080 was added into 1 mL of 10 μM HSA solution, and the mixture was thoroughly mixed by rapid vortices (maintaining a 1:1 molar ratio of protein and dyes). The mixture was then reacted under the shaker at 60 °C for 2 h to obtain the HSA@CO-1080 probe. It was important to note that the above-obtained HSA@CO-1080 probe should be concentrated to the desired concentration for in vivo applications using ultrafiltration concentration. Although not necessary, the protein@dye probes could be further purified with Amicon Centrifugal Filter (10−50 kDa) five times against PBS buffer.

### Synthesis of multicolor imaging probes
HSA@Cy5, HSA@IR-808, β-LG@IR-780, and DIII@CO-1080 probes were synthesized in a similar protocol with the HSA@CO-1080 probe, with the only difference being that the reaction temperatures were adjusted to 37 °C, 60 °C, 70 °C, and 60 °C, respectively. Notably, the RENPs (NaYbF$_4$:Ce, Er@NaYF$_4$:Gd, Yb@PAA) were synthesized based on the previous reports[69,70].

### Quantum yields of different probes
The quantum yields of free CO-1080 in DMSO and HSA@CO-1080 FPs were determined using the dye IR-26 (Φf = 0.5% in 1,2-dichloroehane (DCE)) as a reference. Briefly, a gradient dilution was performed for IR-26, CO-1080, and HSA@CO-1080 in different solvents, ensuring that the OD at 1064 nm was less than 0.1. The corresponding fluorescence intensity was measured under excitation at 1064 nm (> 1100 nm). The integrated fluorescence intensity was plotted as a function of the OD value at 1064 nm and fitted into a linear function. The slopes of the linear fit for IR-26 in DCE, CO-1080 in DMSO, and HSA@CO-1080 FPs in PBS were obtained. The quantum yields of CO-1080 in DMSO and HSA@CO-1080 FPs in PBS were calculated using the following equation:

$$QY_{sample} = QY_{ref} \times \left(\frac{\eta_{sample}}{\eta_{ref}}\right)^2 \times \frac{Slope_{sample}}{Slope_{ref}} \qquad (1)$$

where $QY_{sample}$ is the quantum yield of CO-1080 in DMSO or HSA@CO-1080 FPs in PBS, $QY_{ref}$ is the quantum yield of IR-26 in DCE, $\eta_{ref}$ and $\eta_{sample}$ are the refractive indices of corresponding solvents. $Slope_{sample}$ and $Slope_{ref}$ are the slopes obtained by linear fitting of the integrated fluorescence intensity of IR-26 in DCE, CO-1080 in DMSO, and HSA@CO-1080 FPs in PBS.

### Analysis of protein binding sites
The pure protein or protein@dye samples were first subjected to SDS-PAGE gel electrophoresis, followed by slicing the corresponding gel bands and digesting with trypsin and chymotrypsin. The processed samples were then analyzed by liquid chromatography-mass spectrometry (LC-MS/MS) and the raw files of the original results were obtained. Finally, the data was analyzed and matched using software such as Byonic to obtain the identification results.

### Shotgun proteomics technique
For digestion, DTT solution was added to a final concentration of 10 mmol/L, and reduced in a 56 °C water bath for 1 h. Then, IAM solution was added to a final concentration of 50 mmol/L for 40 min at room temperature in darkness. Next, the enzyme was added with an enzyme-to-substrate mass ratio of 1:100, and the first digestion was performed at 37 °C for 4 h, followed by another enzyme addition at the mass ratio of 1:100 and digested overnight. After digestion, the peptide was desalted using a self-priming desalting column, and the solvent was evaporated in a vacuum centrifuge at 45 °C for the following experiments. Peptide separation was performed using a high-performance liquid chromatography system, with mobile phase A consisting of 0.1% formic acid in water and mobile B consisting of 0.1% formic acid and 80% acetonitrile. The total flow rate was 600 nL/min. The resulting peptides were detected by Q Exactive hybrid quadrupole-Orbitrap mass spectrometer. Raw MS files were analyzed and searched against the target protein database based on the species of the samples using Byonic. The parameters were set as follows: the protein modifications were carbamidomethylation (C) (variable), oxidation (M) (variable), Acetyl (Protein N-term) (variable), CO-1080 (C) (variable); the enzyme specificity was set to trypsin or chymotrypsin; the maximum missed cleavages were set to 3; the precursor ion mass tolerance was set to 20 ppm, and MS/MS mass error was 0.02 Da for fragment ions. Only highly confident identified peptides were chosen for downstream protein identification analysis.

### Hemolysis test
Firstly, approximately 0.5 mL of whole blood was collected from the periocular venous plexus of Balb/c mice. The mice's blood was subsequently centrifuged at 1409 g for 15 min to separate plasma and erythrocytes. This process was repeated 3 times to ensure that the plasma was completely washed off, resulting in the isolation of erythrocytes. The obtained erythrocytes were then diluted with PBS solution to obtain 2% cell dispersion. Next, 100 μL of erythrocyte dispersions were mixed with equal volumes of probes with different

concentrations, and incubated for 1 h in a 37 °C shaker. After incubation, the mixture was centrifuged again at 1409 g for 15 minutes to observe the changes in the supernatant. Finally, the supernatant was taken and the erythrocyte absorption at 577 nm was detected by a microplate reader (Bio-Tek, Synergy LX, USA) to assess the hemolytic effect of the different probes.

### Cell lines and cell culture

Mouse breast cancer cell line (4T1) was obtained from the Shanghai Enzyme Research Biotechnology Co., LTD (Shanghai, China). Mouse fibroblasts cell line (L929) was kindly provided by the Joint Laboratory of Opto-Functional Theranostics in Medicine and Chemistry, First Hospital of Jilin University. All cell lines were cultured in DMEM mixed media with 10% (v/v) fetal bovine serum, 80 U/mL penicillin, and 0.08 mg/mL streptomycin in a relatively humidified atmosphere of 5% $CO_2$ at 37 °C.

### Cytotoxicity evaluation

The cytotoxicity of different probes on L929 and 4T1 cells was detected by MTT assay. Both cell lines were seeded in 96-well plates at a cell density of $1 \times 10^4$ cells/well, and were incubated until they reached approximately 80% confluence. Subsequently, DMEM mixed medium with probe concentrations of 0, 5, 10, 30, 50, 100, and 200 µM was added to the wells and incubated for 24 h. Following removal of the medium in the well plates, MTT (200 µL, 0.5 mg/mL) was added to each well and incubated for 4 h. Finally, the absorbance at 490 nm was measured by a microplate reader (Bio-Tek, Synergy LX, USA). The cytotoxicity of each probe was evaluated according to its absorption values.

### Animal models

All animal experiments were conducted according to the protocols approved by the Animal Ethical Committee of The First Hospital of Jilin University (Procedure Number: 20210642). Balb/c (female, 6–8 weeks), C57 mice (female, 6-8 weeks), SD rats (male, 250–300 g), and Japanese big-ear rabbits (female, 1.5–2 Kg) were purchased from Liaoning Changsheng Biotechnology Co., Ltd and Beijing Vital River Laboratory Animal Technology Co., Ltd. Bedding, nesting material, food, and water were provided ad libitum, and changed and replenished as required. The feeding environment was 20-22 °C, 35–45% humidity, 12 h light-dark alternation.

For the modified perforator flap model, the SD rats (male, 250–300 g) were first shaved and a rectangular flap pedicled with the bilateral inferior gluteal artery (IGA) vessels of approximately 4 cm × 12 cm in dimension was designed over the central dorsum of the rat. Then the flap was disinfected, and flap dissection was performed at the subcutaneous deep fascia of the central dorsum of the rat. All aortic vessels including deep circumflex iliac artery (DCI), posterior intercostal artery (PIC), and thoracodorsal artery (TD), in the dissecting path, were ligated and eventually exposed to bilateral IGA perforators. Finally, the flap was sutured in situ using a 3-0 nonabsorbable suture.

For the tumor lymph node metastasis model, $5 \times 10^6$ breast cancer cells (4T1) were suspended in 100 µL PBS and injected subcutaneously into the dorsum of each Balb/c mice (female, 6–8 weeks). Then, 30 days after tumor inoculation, the mice were applied for tumor lymph node metastasis experiments. The maximal tumor size permitted by the ethics committee is 1000 mm³, and the maximal tumor size used in this work was not exceeded.

### NIR-I/II imaging

NIR-I/II imaging set-up was built in-house and consisted of a camera (Princeton Instruments, NIRvana-640), laser (Artemis Intelligent Imaging), and off-the-shelf optics (Thorlabs, Edmund optics, etc.). The excitation light was generated by a 660 nm, 808 nm, 980 nm, or 1064 nm fiber-coupled diode laser with adjustable power density. Different long-pass (LP) filters were combined to capture different waveband images in the NIR-II window. All mice were shaved with depilatory cream and anesthetized with isoflurane before imaging. During imaging experiments, at least 3 mice were used as parallel controls. All fluorescence images were processed and analyzed by Image J software.

### Statistical analyzes

Data points were collected and compiled in Microsoft Excel, and statistical analyzes were performed using GraphPad Prism and Origin Pro software. Statistical significance was determined by a two-tailed Student's t-test, with a p-value of less than 0.05 considered significant. For the continuous variables, the data were presented as mean ± standard deviation.

### Reporting summary

Further information on research design is available in the Nature Portfolio Reporting Summary linked to this article.

## Data availability

The authors declare that the source data supporting the findings of this study are available within the article and its Supplementary Information files. Due to the substantial size of the raw imaging data, they are available upon request from the corresponding author and will be provided within two weeks of the request being made. Source data are provided with this paper.

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

## Acknowledgements

This work was supported by the National Key Research and Development Program of China (2022YFC2408100).

## Author contributions

J.X. and S.Z. designed the experiments. J.X. was responsible for the design, synthesis, and characterization of the probes, as well as manuscript preparation. N.Z. and T.H. contributed to relevant dye characterization. Y.D., X.Z. and J.L. assisted with part of bioimaging and animal model construction. J.X. and S.Z. discussed the results and drafted the manuscript. All authors contributed to the proofreading of the manuscript.

## Competing interests

The authors declare no competing interests.
