## [Peer Review File · Nature Communications]

Reviewers' Comments:

Reviewer #1:

Remarks to the Author:

This manuscript develops a protein-seeking strategy to construct a series of biomimetic NIR-II fluorescent proteins (NIR-II FPs) used as probes for high-performance NIR-II lymphography and angiography. Differentiating from traditional gene-coding fluorescence proteins, chemogenic protein-seeking NIR-II dye (CO-1080) as the chromophore automatically covalently tag to the structure-appropriate proteins (e.g., HSA) via a nucleophilic substitution reaction to form the NIR-II FPs. And the authors explore the molecular mechanism underlying the interaction between proteins and dye molecules in detail. A multichannel imaging system is achieved providing for integrated monitoring of various biological events with longer emission wavelengths and without crosstalk. Overall, the manuscript is innovative in method and meaningful, containing a wealth of experimental data, but there are some issues need addressing before accepting it for publication:

1. Expect for β -Lactoglobulin (β -LG), bovine serum albumin (BSA), and human serum albumin (HSA), did the authors attempt to use other proteins containing pocket with -SH groups as protein shells to construct NIR-II FPs?
2. The authors selected L-cysteine hydrochloride molecules as Cys residue substitutes for HSA, and validate covalent formation via displacement of chloride on 1080 dyes by thiol group on Cys residues under the base catalyst conditions, but why HSA could react with 1080 dyes without catalysis?
3. How to understand the difference to analyze the interaction between CO-1080 and HSA by glide docking, covalent docking, and dynamic simulation. Explain which simulation method you think is closest to the real situation and give the corresponding reasons.
4. In Figure 5, How HSA @ CO-1080 FPs were injected into mice to achieve NIR-II lymphography and angiography and what is the exposure time of imaging? And in Figure 5f, what is the signal-to-noise ratio of angiography?
5. In Supplementary Fig. 40, the fluorescence intensity of DII@CO-1080 was weaker than HSA@CO-1080, Have the authors explored the reasons for this phenomenon?

Reviewer #2:

Remarks to the Author:

This study reports a chemogenic protein-seeking strategy for fabricating biomimetic near-infrared-II fluorescent proteins by covalently linking dyes and specific binding sites in proteins. The interaction mechanism between proteins and NIR-II chromophores was systematically investigated. The obtained HSA@CO-1080 FPs enabled integrated two-color NIR-II imaging of various biological events without interference, such as lymph node colocalization, lymphatic-vascular visualization, and pharmacokinetics assessment. However, I feel the novelty of this work is moderate. Actually, the author has published similar studies recently (Tian et al., *Sci. Adv.* 2019; 5, eaaw0672; Xu et al., <https://doi.org/10.1002/adhm.202301051>). Thus, the concepts of protein-seeking dyes and NIR-II FPs lack innovation. In addition, the proteins and dyes used here are not novel. In conclusion, the paper is well written, but not stands at the level of *Nat. Comm.*

1. The encapsulation of NIR-I/II fluorophores with proteins has been proven as an effective approach to enhance their brightness and pharmacokinetics. Although the authors have made great efforts to identify the specific covalent binding site between HSA and NIR-II chromophore, a consistent binding pattern between HSA protein and chromophores was not revealed. For example, why do Et-1080, FD-1080, and CO-1080 possess different affinities with HSA? What changes should be made in order to enhance the binding energy? Also, the detection of binding behavior after HSA Cys477 mutation might be more rigorous to identify binding sites.
2. The authors identified the DIII domain as the binding domain between HSA and CO-1080. But the brightness of HSA@CO-1080 complex was higher than DIII@CO-1080. What's the reason? The authors claimed the binding fractions of DIII to Et-1080, FD-1080, and CO-1080 were 34.34%, 15.12%, and 100%, respectively. But is this calculated from high-resolution mass spectrometry? The HPLC data is required to determine the accurate content of the protein-chromophore complex.
3. The authors have conducted lymphography and angiography with HSA@CO-1080, just like the applications in their previous studies. In this manuscript, they also focus on multicolor in-vivo

imaging using diverse protein-chromophore complexes. How about the biosafety for IR-808 and IR-780? Additionally, the potential clinical applications of multicolor imaging should be discussed.

4. To validate the feasibility of the NIR-I/II FPs for multicolor bioimaging, the authors selected NIR-I protein-seeking dyes to construct biomimetic NIR-I FPs (HSA@IR-808 and β -LG@IR-780). What is the basis for selecting different protein shells? Please explain the binding mechanism of β -LG with IR-780?

5. In Figure 5, HSA@CO-1080 FPs demonstrated superior systemic and local (leg) vascular NIR-II imaging capabilities compared to the CO-1080 chromophores. The blood half-life of HSA@CO-1080 FPs should be described in this section, which is essential for angiography.

6. The encapsulation of NIR-I/II fluorophores with protein has been widely used as an effective approach to biomedical imaging and imaging-guided photothermal therapy (DOI: 10.1002/adhm.201800589). The obtained complexes also possess the improved biocompatibility and enhanced luminescence efficiency. In this manuscript, the authors fabricated the biomimetic near-infrared-II fluorescent proteins by covalently linking dyes CO-1080 with specific binding sites in proteins of HSA. This method lacks generalizability that could be generalized to other proteins and NIR-II chromophores. What is the advantage of biomimetic fluorescent proteins compared to the protein/dye complexes mentioned above?

7. What is the quantum yield of the CO-1080 dye and HSA@CO-1080 FPs?

8. What is the quality of in vivo fluorescence imaging with HSA@CO-1080 FPs? e.g., the depth of imaging, imaging resolution, signal-to-noise ratio?

Reviewer #3:

Remarks to the Author:

Dear authors,

I have read carefully the manuscript submitted for consideration to NC.

The authors synthesized a new conjugate of cyanine dye and HSA. The topic is of interest. However, the conclusions are not supported by the data presented.

I143.

The binding with protein at high temperature can affect the binding ability. Have you performed the experiments dye-protein noncovalent binding, like measuring binding constants with native HSA and heated?

I193.

a. Please provide excitation and emission spectra of dyes and their complexes with protein, both covalent and non-covalent complexes.

b. The J-aggregate formation is also possible in regarded conditions. It is necessary to verify that observed fluorescence is not emitted by aggregates.

I215-I217.

According to proposed binding mechanism, the dyes have identical chromophore structure. Thus it is unobvious the CO-1080 binds to HSA most effectively, because of steric hindrance.

I230.

Comparison of dyes non-covalent binding with protein and binding constants need to be provided. The fluorescence lifetime for each dye and both types of complexes can significantly improve manuscript.

I234.

The site-specific binding with FBZ and Ibuprofen can confirm the localisation in DIII site. It can support the docking information.

I407.

The absence of photodynamic activity need to be proven. The triplet state formation and singlet oxygen generation in nonpolar organic solvents and PBS with Dye-HSA complex.

I411.

Need to be compared with using CO-1080 separately.

Minor:

line 60 and 61 chromophobe need to be replaced with chromophore
l193.

What means PL intensity on graph? Fluorescence?

Point by point replies to reviewers and revisions made.

Reviewer #1

This manuscript develops a protein-seeking strategy to construct a series of biomimetic NIR-II fluorescent proteins (NIR-II FPs) used as probes for high-performance NIR-II lymphography and angiography. Differentiating from traditional gene-coding fluorescence proteins, chemogenic protein-seeking NIR-II dye (CO-1080) as the chromophore automatically covalently tag to the structure-appropriate proteins (e.g., HSA) via a nucleophilic substitution reaction to form the NIR-II FPs. And the authors explore the molecular mechanism underlying the interaction between proteins and dye molecules in detail. A multichannel imaging system is achieved providing for integrated monitoring of various biological events with longer emission wavelengths and without crosstalk. Overall, the manuscript is innovative in method and meaningful, containing a wealth of experimental data, but there are some issues need addressing before accepting it for publication:

Response:

We thank the reviewer for recognizing the novelty of our work. We also highly appreciate the reviewer's suggestions for strengthening our work. Here, we listed a detailed response to the reviewer's comments.

1. Expect for β -Lactoglobulin (β -LG), bovine serum albumin (BSA), and human serum albumin (HSA), did the authors attempt to use other proteins containing pocket with -SH groups as protein shells to construct NIR-II FPs?

Response and revisions: Thanks for highlighting this important question. In fact, in addition to the β -lactoglobulin (β -LG), bovine serum albumin (BSA) and human serum albumin (HSA) mentioned in this manuscript, we have also attempted to construct NIR-II FPs using other pocket proteins with -SH groups as protein shells, including lactotransferrin (Lf), ovalbumin (OVA), α -lactalbumin (α -La), and others (**Figure R1** and **Figure R2**). Unfortunately, these proteins did not exhibit preferable covalent binding and fluorescence enhancement to CO-1080 dyes, resulting in lower production efficiency for NIR-II FPs. Therefore, we only included the proteins with higher binding efficiency to the NIR-II chromophore (CO-1080) as outer shells to construct biomimetic fluorescent proteins in this manuscript. For the optimal proteins (e.g., HSA), the displacement of meso-chlorine between CO-1080 and HSA could be completed under mild PBS buffer conditions, and our protein-seeking approach explained that the DIII domain of the HSA acted as a microreactor, triggering a catalysis-free nucleophilic substitution by efficiently adjusting the CO-1080 dye conformation.

Figure R1. Schematic of the reaction of different proteins with CO-1080 dye, including β -lactoglobulin (β -LG), bovine serum albumin (BSA), human serum albumin (HSA), α -lactalbumin (α -La), ovalbumin (OVA), and lactotransferrin (Lf).

Figure R2. (a) Fluorescence intensity of different proteins reacted with CO-1080 dye at different temperatures. (b) SDS-PAGE gel electrophoresis of different proteins reacted with CO-1080 dye at different temperatures.

2. The authors selected L-cysteine hydrochloride molecules as Cys residue substitutes for HSA, and validate covalent formation via displacement of chloride on 1080 dyes by thiol group on Cys residues under the base catalyst conditions, but why HSA could react with 1080 dyes without catalysis?

Response and revisions: Thank you for this valuable feedback. In order to confirm the covalent reaction site between protein and dye, we first selected the L-cysteine hydrochloride molecule as a Cys residue substitute for HSA to react with the CO-1080 dye. Unfortunately, as shown in **Figure 4g**, the L-cysteine hydrochloride molecules and CO-1080 dye did not react with the CO-1080 dye under the same reaction conditions as the HSA. Therefore, after continuous attempts, we found that L-cysteine hydrochloride molecules could replace the chloride on 1080 dyes with thiol group on Cys residues under the alkaline catalyst conditions, which confirmed that the covalent bonding site between protein and dye was Cys residue.

As to why HSA could react with 1080 dyes without catalysis, in addition to the small molecule experiments mentioned above, we have also tried further experiments with short peptides containing Cys residues (glutathione). As shown in **Figure R3**, the high-resolution mass spectrometry (MALDI-TOF-MS) data showed that the glutathione did also not covalently bind with CO-1080 dye without catalysis. Collectively, we concluded that HSA, with its unique three-dimensional helical hydrophobic cavity structure, could function as a microreactor to trigger a catalysis-free nucleophilic substitution by efficiently adjusting the CO-1080 dye conformation. However, for CO-1080 dyes, small molecules and polypeptides could hardly restrict their free movement and conformations. Therefore, in the absence of catalysts, it was difficult for CO-1080 dyes to effectively collide with either L-cysteine hydrochloride molecule or glutathione to achieve nucleophilic substitution reactions.

Figure R3. (a) Schematic diagram of the bonding between CO-1080 and glutathione (GSH). (b) MALDI-TOF-MS results before and after binding between CO-1080 and glutathione (GSH).

3. How to understand the difference to analyze the interaction between CO-1080 and HSA by glide docking, covalent docking, and dynamic simulation. Explain which simulation method you think is closest to the real situation and give the corresponding reasons.

Response and revisions: Thank you for this valuable feedback. Indeed, we have adopted different simulation methods including glide docking, covalent docking, and dynamic simulation to analyze the interaction between CO-1080 and proteins. For glide docking simulation, it belonged to non-covalent binding simulation, which is also a common simulation method to study the interaction between proteins and ligands. In this manuscript, it was mainly used to analyze the binding sites

and binding ability between proteins and dyes. For covalent docking simulation, the main focus is on the specific covalent bond interactions between proteins and the dyes. That is, after determining the specific covalent binding sites based on proteomic analysis, the precise site covalent docking simulation was implemented between the protein and the dye. For dynamic simulation, it was a further conformational optimization after covalent docking simulation between protein and dye. After careful analysis of the covalent docking simulation results between protein and dye, we found that the conformation between the two was not optimal. Therefore, in view of this fact, we carried out further dynamic simulation optimization to determine the optimal stable conformation between protein and dye.

Overall, it could be understood that the above three simulation processes (glide docking, covalent docking, and dynamic simulation) correspond to the three successive stages of protein-dye interaction: In stage I, the protein-seeking CO-1080 dye inserts into the hydrophobic pocket of HSA and binds to the pocket through supramolecular interactions. In stage II, a covalent bond is formed by the -SH group of Cys477 and the Cl-C group of the CO-1080 dye through nucleophilic substitution under the restricted microreactor of HSA. In stage III, the covalent conformation between CO-1080 dye and HSA undergo fine-tuning to achieve the final steady-state bright-emitting NIR-II protein@dye complex. Therefore, we considered that the optimal stable conformation between protein and dye obtained by dynamic simulation was the closest to the real situation.

4. In Figure 5, How HSA@CO-1080 FPs were injected into mice to achieve NIR-II lymphography and angiography and what is the exposure time of imaging? And in Figure 5f, what is the signal-to-noise ratio of angiography?

Response and revisions: We sincerely appreciate the reviewer for bringing this issue to our attention. The experimental details have been refined in the revised manuscript. For NIR-II lymphography, the HSA@CO-1080 FPs were injected into mice by intramuscular injection. For NIR-II angiography, the HSA@CO-1080 FPs were injected into mice via tail intravenous injection. The exposure time of the corresponding imaging has also been marked in the corresponding locations in the revised manuscript. Besides, we have also added the signal-to-noise ratio of angiography in **Figure 5d**.

5. In Supplementary Fig. 40, the fluorescence intensity of DIII@CO-1080 was weaker than HSA@CO-1080, Have the authors explored the reasons for this phenomenon?

Response and revisions: Thank you for this valuable feedback. It was well known that DIII belonged to one of the HSA protein domains and was adjacent to the DII domain. By carefully comparing the theoretical simulation results of HSA and CO-1080, it was found that CO-1080 dyes were mainly confined in the DIII domain, but a small part of them were exposed to the DII domain (**Figure R4a**). Therefore, as shown in **Figure R4a-c**, dye molecules had a certain degree of exposure when CO-1080 binds to DIII compared to intact proteins (HSA), which also indirectly reduced the domain-limiting ability of protein molecules to bind with dye molecules, thereby resulting in a decrease in docking score and binding energy between protein and dye. Thus, compared with HSA, the fluorescence enhancement effect of DIII on CO-1080 dyes were weakened. Meanwhile, the partial exposure of dye molecules also increases the possibility of contact quenching between molecules. Overall, these analysis results effectively explain why the fluorescence intensity of DIII@CO-1080 was weaker than HSA@CO-1080.

Figure R4. (a) Theoretical simulation of CO-1080 dye binding to HSA and DIII proteins by gliding docking mode. Comparison of (b) docking scores and (c) binding energy of CO-1080 with HSA and DIII proteins.

Reviewer #2

This study reports a chemogenic protein-seeking strategy for fabricating biomimetic near-infrared-II fluorescent proteins by covalently linking dyes and specific binding sites in proteins. The interaction mechanism between proteins and NIR-II chromophores was systematically investigated. The obtained HSA@CO-1080 FPs enabled integrated two-color NIR-II imaging of various biological events without interference, such as lymph node colocalization, lymphatic-vascular visualization, and pharmacokinetics assessment. However, I feel the novelty of this work is moderate. The author has published similar studies recently (Tian et al., *Sci. Adv.* 2019; 5, eaaw0672; Xu et al., <https://doi.org/10.1002/adhm.202301051>). Thus, the concepts of protein-seeking dyes and NIR-II FPs lack innovation. In addition, the proteins and dyes used here are not novel. In conclusion, the paper is well written, but not stands at the level of *Nat. Comm.*

Response: We would like to express our gratitude to the reviewers for taking the time out of their busy schedules to review our work and provide us with constructive comments. Even though the concept of covalent binding between proteins and Cl-containing cyanine dyes has been reported in our earlier studies, this work focuses on studying NIR-II probes and their imaging applications from the perspective of biomimetic fluorescent proteins. While the encapsulation of NIR-I/II fluorophores with proteins, whether exogenous or endogenous, has been proven as an effective approach to enhance their brightness and pharmacokinetics, the molecular mechanism underlying the interaction between proteins and dye molecules has been poorly understood and typically referred to as “complexes”. It is important to emphasize that there are significant differences between this study and the previous findings mentioned above. I would like to highlight several novel aspects of this work:

1. Regarding the binding behavior, there were few reports on the protein-dye complexes exhibiting explicit covalent binding behavior. Previously reported protein-dye complexes were mostly constructed using protein encapsulation strategies that relied on simple physical encapsulation (*Adv. Healthcare Mater.* 2018, 1800589; *Adv. Funct. Mater.* 2019, 1906343; *ACS Appl. Bio Mater.* 2020, 3, 9126–9134), and the specific binding mechanism between protein and dye molecules remained unclear. On the contrary, the near-infrared-II fluorescent proteins constructed in this work has been successfully elucidated by multi-directional experiments, demonstrating precise and controllable covalent binding behavior and a specific binding mechanism. In comparison to traditional physical encapsulation strategy, the precise and controllable covalent binding between protein and dye with an equimolar stoichiometric ratio could also effectively avoid the intrinsic aggregation-induced quenching (ACQ) phenomenon of traditional dyes and the biosafety concerns caused by dye leakage. Notably, the current chemical modifications of dyes (such as NHS-modified, Mal-modified, azide modified, etc) to achieve covalent bonding with proteins suffered from random dye conjugation, limited brightness enhancement, and potential influence on protein activity. Conversely, our strategy achieves high-efficient and controllable covalent binding between proteins and dyes under very mild physical conditions, resulting in biomimetic fluorescent proteins with significantly enhanced brightness and stability.
2. In terms of dye molecules, there have been no reported NIR-II peak emission dyes with a clearly defined covalently bound protein mechanism. Our previous work has investigated the covalent bonding behavior between protein and dye molecules. However, these studies exclusively

focused on NIR-I cyanine dyes and can only achieve the NIR-II imaging using the weak tail emission. Instead, the current work relies on NIR-II cyanine dyes to achieve highly efficient covalent binding with proteins, effectively overcoming the limitations of previous studies. Furthermore, the systematic exploration of the covalent binding behavior between NIR-II cyanine dyes and proteins study will serve as a valuable reference and inspiration for the development of higher-performance NIR-II dyes that can covalently bind with proteins. It is worth emphasizing that we are actively engaged in the design and development of new dye molecules to expand the library of NIR-II dyes that can be covalently matched with a large number of proteins.

3. In previous studies, it has been demonstrated that protein shells can covalently bind to cyanine dyes, effectively enhancing their luminescent properties. However, there is currently a shortage of protein libraries with high-efficiency covalent binding for cyanine dyes. Therefore, it is crucial to intensify efforts to screen out more proteins that exhibit high gain effect on NIR cyanine dyes. Here, we compared and analyzed the effects of different protein shells and different structural dye on the binding behavior between proteins and NIR-II dye molecules from multiple perspectives, providing effective technical background and inspiration for future research.
4. As for the field of fluorescent proteins (FPs), the current gene-edited NIR-I FPs cannot be scaled up for production under limited lab or industry conditions, and there are very few NIR-II FPs with emissions > 1000 nm available. To address these challenges, we propose a novel biomimetic approach to construct highly-bright, photostable, and biocompatible NIR-I/II fluorescent proteins (**Scheme 1**). Herein, the main breakthroughs achieved in this work were summarized again as follows:

Firstly, our chemogenic protein-seeking approach was universal and applicable to creating a wide range of biomimetic NIR-I/II FPs. It allowed for the utilization of proteins with favorable pocket conformation/thiol groups and dyes with appropriate structures to create spectrally separated multicolor NIR-I/II FPs, thereby facilitating the integrated visualization of multiple biological events in real time without crosstalk.

Secondly, chemically generated protein-seeking methods allow for the large-scale generation of NIR-I/II fluorescent proteins under very mild physiological conditions without the need for catalysis. Notably, it was also able to form NIR-I/II FP for deep tissue imaging in situ in vivo. This simple and effective bionic approach has the potential to revolutionize biometrics-based fluorescent proteins and maximize the clinical translational benefit of NIR-II fluorescence imaging techniques.

Thirdly, the biomimetic NIR-II fluorescent proteins enable high-performance NIR-II lymphography and angiography, such as image-guided precise dissection of lymph nodes, high-quality visualization of lymphedema diseases, identifying vascular lesions in various organs, and evaluating postoperative blood flow recovery. Notably, our NIR-II fluorescent proteins provide a noninvasive and effective method of monitoring flap functionality, with potential for human use.

Finally, our study also provided a comprehensive exploration of the formation mechanism of NIR-II fluorescent proteins (FPs) using proteomics analysis and docking modeling. We found that the hydrophobic cavity of specific protein tags acted as a microreactor, promoting the formation of FPs under mild physiological conditions. Specifically, for the typical NIR-II FPs

HSA@CO-1080, we identified Cys 477 on DIII of HSA as the specific binding site for CO-1080. This discovery provides new insights into the fundamental interaction principles of biomolecules and lays a foundation for the future development of novel biomimetic NIR-I/II FPs.

Scheme 1.

Overall, our study breaks through the traditional genetically encoded methods to construct NIR fluorescent proteins and provides a fundamental principle for designing new dyes for in-situ protein targeting purposes and more high-performance NIR-II protein-dye covalent complexes.

1. The encapsulation of NIR-I/II fluorophores with proteins have been proven as an effective approach to enhance their brightness and pharmacokinetics. Although the authors have made great efforts to identify the specific covalent binding site between HSA and NIR-II chromophore, a consistent binding pattern between HSA protein and chromophores was not revealed. For example, why do Et-1080, FD-1080, and CO-1080 possess different affinities with HSA? What changes should be made in order to enhance the binding energy? Also, the detection of binding behavior after HSA Cys477 mutation might be more rigorous to identify binding sites.

Response and revisions: Thank you for this valuable feedback. Indeed, the covalent binding behavior between HSA and NIR-II chromophores was systematically investigated in this manuscript using theoretical simulations, high-resolution mass spectrometry, proteomics, and other experiments. Among them, as shown in **Figure 2**, the CO-1080 dye was finally determined to be the most compatible chromophore with HSA by optimizing the binding behavior between HSA and different chromophore (Et-1080, FD-1080, and CO-1080). In fact, through theoretical simulation results, it could be clearly found that the difference of side chains between different chromophores directly affected its binding behavior with HSA, and indirectly led to the difference of optical properties of the constructed the protein@cyanine dye probes. The different side groups determine the

hydrophilicity/hydrophobicity of dyes themselves, and then affect the interaction between proteins and different dyes. Theoretical simulation results (**Figure 2f-j**) also clearly showed that the CO-1080 dye had the largest number of amino acid interactions with HSA, as well as the highest docking score and binding energy, surpassing the affinity of Et-1080 and FD-1080 dyes.

For the cyanine dye structure, it mainly consists of the end-group head, linker, and side chain. Therefore, it could be predicted that the changes in the above three parts may affect the binding energy between protein and dye. During this round of revision, we conducted additional sets of experiments to confirm the different affinities between dye and protein through synthesizing several sets of dyes. First, as shown in **Figure R5**, when comparing the binding behavior of different NIR-II dyes and HSA, it was evident that the differences in the end-group heads directly impact the fluorescence enhancement and covalent binding between proteins and dyes, primarily due to steric hindrance. Secondly, according to **Figure R6** and **Figure R7**, it could be effectively confirmed that the side chain could directly affect the binding behavior between the protein and the dye. As shown in **Figure R6**, with the increase of side chain length, the NIR-II brightness and covalent bond efficiency of HSA@CO-1080-Cn series increased first and then decreased, and CO-1080-C6 (CO-1080) and HSA exhibited the best bonding behavior. Meanwhile, by comparing the 1080 series dyes with different side chain end groups, it was also clearly found that different end groups significantly affected the binding behavior (**Figure R7**). The outstanding binding behavior between 1080 series dyes with carboxyl end group (CO-1080) and protein was obviously better than that of 1080 series dyes with alkyl group (Et-1080-C6). Besides, the effects of different linkers on the binding of dyes to proteins were also investigated in **Figure R8**. Results showed that different linkers directly affected the binding behavior, and IR-780 (cyclohexene structural linker) showed efficient covalent binding with HSA. Collectively, the binding energy between protein and dye could be effectively enhanced by adjusting the structure of chlorinated cyanine dyes, including the conjugated structure of the end-group head, the length of the side chain, the side chain end groups, the linker structure, etc.

Notably, this reviewer provided a very constructive comment suggesting the use of Cys477-mutated HSA to identify binding sites. We conducted numerous mutation experiments and found that such mutated protein was not stable enough. However, we still successfully obtained four variants of albumin, where Cys residues at positions 477, 487, and 34 were selectively mutated to Gly (as shown in **Figure R9**). We included the Cys487 mutation as it has been detected in our proteomics results and they were spatially adjacent. Although the key recombinant HSA with both Cys477 and Cys487 mutations degraded for unknown reasons, we still obtained an alternative recombinant HSA with Cys477, Cys487, and Cys34 mutations.

These mutated albumin variants were then reacted with CO-1080 at the optimal temperature of 60°C, it was observed that when only one of the residues at positions Cys477 or Cys487 was mutated, CO-1080 could still form a covalent bond with the mutated albumins. This indicated that even though Cys477 was confirmed as the optimal reaction site by proteomics, Cys487 and other spatially adjacent cysteine (Cys461, Cys476) were also potential reactive sites in the absence of Cys477. In contrast, when both residues were mutated, the ability for covalent binding significantly decreases, resulting in nearly no dye-labeled band. This finding supported our mass spectrometry and proteomics results.

[Redacted]

Figure R5. (a) Molecular structure of different NIR-II dyes, including IR-1048, IR-1061, IR-26, Et-1080, CO-1080, and FD-1080. (b) NIR-II brightness of different NIR-II dyes in DMSO, PBS, and HSA. (c) Comparison of NIR-II brightness after binding between HSA and different NIR-II dyes. (d) SDS-PAGE gel electrophoresis after binding between HSA and different NIR-II dyes.

[Redacted]

Figure R6. (a) Reaction schematic of the HSA and CO-1080 dyes with different side chain lengths. (b) NIR-II brightness of the CO-1080-C_n in PBS and HSA@CO-1080-C_n. (c) SDS-PAGE gel electrophoresis of the CO-1080-C_n in PBS and HSA@CO-1080-C_n.

Figure R7. (a) Molecular structure of the Et-1080-C6 and CO-1080 dyes. (b) NIR-II brightness of Et-1080-C6 and CO-1080 in PBS and HSA. (b) SDS-PAGE gel electrophoresis of HSA@Et-1080-C6 and HSA@CO-1080.

Figure R8. (a) Molecular structure of dyes with different linkers. (b) NIR-II brightness of different linker dyes in PBS and HSA. (c) SDS-PAGE gel electrophoresis after binding of different linker dyes and HSA.

[Redacted]

Figure R9. (a) DNA sequencing results of different mutant proteins, including rHSA-C477G, rHSA-C487G, rHSA-C477G-C487G, rHSA-C34G-C477G-C487G, and rHSA. (b) SDS-PAGE gel electrophoresis of different mutant proteins (Left) and different mutant protein@CO-1080 complexes (Right).

2. The authors identified the DIII domain as the binding domain between HSA and CO-1080. But the brightness of HSA@CO-1080 complex was higher than DIII@CO-1080. What's the reason? The authors claimed the binding fractions of DIII to Et-1080, FD-1080, and CO-1080 were 34.34%, 15.12%, and 100%, respectively. But is this calculated from high-resolution mass spectrometry? The HPLC data is required to determine the accurate content of the protein-chromophore complex.

Response and revisions: Thank you for this valuable feedback. DIII is a protein domain belonging to the HSA protein, located adjacent to the DII domain. Through a careful comparison of the theoretical simulation results of HSA and CO-1080, it was observed that CO-1080 dyes were primarily inserted into the DIII domain, with a small portion exposed to the DII domain (**Figure**

R4a). Consequently, as depicted in **Figure R4a-c**, the dye molecules exhibited a certain degree of exposure when binding to DIII, in contrast to intact HSA. This indirectly reduced the protein molecules' ability to confine and limit the dye molecules within the domain, resulting in a decrease in the docking score and binding energy between the protein and dye. As a result, the fluorescence enhancement effect of DIII on CO-1080 dye was weakened compared to HSA. In addition, the partial exposure of the dye molecules' structures also increased the possibility of contact quenching between molecules. In summary, the aforementioned analysis effectively explains why the fluorescence intensity of the HSA@CO-1080 complex was higher than that of the DIII@CO-1080 complex.

Meanwhile, the binding fractions of DIII to Et-1080, FD-1080 and CO-1080 in this manuscript was indeed calculated from the high-resolution mass spectrometry (HRMS). Admittedly, there were also many studies used the HRMS data to quantitatively characterize the binding efficiency and the binding ratio between different molecules (Wakankar, A. *mAbs*, 2011, 3, 161-172; *Anal. Chim. Acta* 955 (2017) 67e78; *Anal. Chem.* 2014, 86 (21):10674-83; *Rapid Commun. Mass Spectrom.* 2005; 19: 1806–1814; Thermo Fisher Scientific (High Resolution Mass Spectrometry of Antibody Drug Conjugates Using the Orbitrap Mass Analyzer). Therefore, based on literature reports and confirmation by high-resolution mass spectrometer technicians, we utilized the relative abundance values (%) of the corresponding sample molecular weights to quantitatively analyze the binding fraction between protein and dye small molecules. As shown in **Figure R10** and **Figure R11**, the molecular weight position relative abundance values of the DIII@CO-1080 complexes were carefully calculated. Specifically, the binding case of DIII and CO-1080 was taken as an example. The sum of the relative abundances of the DIII@CO-1080 complex was then divided by the sum of the relative abundances of both the DIII and DIII@CO-1080 complexes. This calculation yielded the binding fraction between DIII and the CO-1080 dye.

In addition, we also attempted to characterize the accurate content of the protein-chromophore complex using HPLC as suggested by the reviewers. Unfortunately, as shown in **Figure R12** and **Table R1**, due to the small polarity difference between the pure protein (HSA) and the protein-chromophore complex (HSA@Et-1080, HSA@CO-1080, and HSA@FD-1080), the accurate content of the protein-chromophore complex could not be efficiently characterized by HPLC.

Figure R4 (a) Theoretical simulation of CO-1080 dye binding to HSA and DIII proteins by gliding docking mode. Comparison of (b) docking scores and (c) binding energy of CO-1080 with HSA and DIII proteins.

↓

Mass (Da)	Abundance	Relative abundance	Credibility	Charge distribution	Elution time
23569.23	6.15E+09	100.00	48.62	14 - 27	4.05
23369.11	3.96E+09	64.37	49.04	13 - 27	4.03
23731.47	4.04E+08	6.57	39.84	14 - 23	3.98
23586.31	2.85E+08	4.64	45.16	14 - 24	3.93
23528.79	2.50E+08	4.07	40.54	14 - 24	3.98
23350.79	2.39E+08	3.88	45.01	13 - 23	4.17
23550.67	2.30E+08	3.75	41.97	14 - 23	4.17
23440.17	2.04E+08	3.32	29.08	14 - 22	4.03
23893.90	1.08E+08	1.76	40.11	14 - 23	3.93
23526.41	9.26E+07	1.51	33.40	14 - 22	4.13
23855.71	7.55E+07	1.23	42.22	14 - 23	3.90

Figure R10. High-resolution mass spectrogram and specific data analysis chart of DIII.

Mass (Da)	Abundance	Relative abundance	Credibility	Charge distribution	Elution time
24232.01	5.05E+09	100.00	52.55	14 - 24	4.47
24031.79	3.25E+09	64.32	52.90	14 - 24	4.44
24895.09	4.38E+08	8.66	52.98	14 - 24	4.33
24394.25	4.06E+08	8.03	48.27	14 - 23	4.40
24248.64	3.19E+08	6.31	50.86	14 - 24	4.33
24694.81	2.83E+08	5.59	54.12	14 - 23	4.33
24192.33	2.34E+08	4.62	51.43	14 - 23	4.38
24260.33	2.30E+08	4.54	54.18	14 - 23	4.70
23569.42	2.04E+08	4.03	44.70	14 - 23	4.16
24356.56	2.02E+08	3.99	45.75	14 - 23	4.45
24264.24	1.91E+08	3.78	52.32	14 - 24	4.27
24213.53	1.67E+08	3.30	47.10	14 - 23	4.61
24013.47	1.57E+08	3.11	45.86	14 - 23	4.59
24102.96	1.50E+08	2.96	49.41	14 - 23	4.45
24063.83	1.45E+08	2.88	54.26	14 - 24	4.27
24060.09	1.37E+08	2.71	52.56	14 - 23	4.70
23369.10	1.29E+08	2.56	56.10	13 - 23	4.16
24518.28	1.28E+08	2.54	54.83	14 - 23	4.35
24556.30	1.26E+08	2.49	50.68	14 - 23	4.35
24048.40	1.07E+08	2.11	48.29	14 - 23	4.33
24190.45	9.42E+07	1.86	41.24	14 - 23	4.52
24718.55	8.69E+07	1.72	51.74	14 - 23	4.35
23987.53	5.86E+07	1.16	49.30	14 - 23	4.38

Figure R11. High-resolution mass spectrogram and specific data analysis chart of DIII@CO-1080.

Figure R12. HPLC chromatogram of (a) HSA, (b) HSA@Et-1080, (b) HSA@CO-1080, and (d) HSA@FD-1080.

Table R1. Peak area of different sample

Samples	Retention time (min)	Peak area	% area
HSA	10.782	36975	7.46
	11.968	458387	92.54
HSA@Et-1080	10.727	35087	6.26
	11.964	525460	93.74
HSA@CO-1080	10.648	44923	7.55
	11.930	550242	92.45
HSA@FD-1080	10.757	27915	6.67
	11.961	390657	93.33

3. The authors have conducted lymphography and angiography with HSA@CO-1080, just like the applications in their previous studies. In this manuscript, they also focus on multicolor in-vivo imaging using diverse protein-chromophore complexes. How about the biosafety for IR-808 and IR-780? Additionally, the potential clinical applications of multicolor imaging should be discussed.

Response and revisions: Thank you for your valuable feedback. Indeed, the primary objective of the basic performance studies on HSA@CO-1080 FPs in lymphography and angiography is to provide a comparative analysis with previous reports. It is worth noting that NIR-II FPs is a high-performance probe, and our main focus was to explore new applications using such NIR-II FPs, such as rat flap imaging and multicolor imaging. In addition, we also extended the use of the HSA@CO-1080 probe in NIR-II neuroimaging. As depicted in **Figure R13**, the HSA@CO-1080 probes effectively illuminated nerves, demonstrating superior NIR-II neuroimaging capabilities compared to the pure CO-1080 dye. Importantly, the HSA@CO-1080 probe successfully achieved accurate localization and tracking of nerve injuries, as shown in **Figure R14**.

Meanwhile, we also investigated the biosafety of IR-780, IR-808, and their corresponding protein-chromophore complexes based on the feedback from the reviewers. As depicted in **Figure R15**, the cytotoxicity data demonstrated that there was no observable cytotoxicity for β -LG@IR-780 or HSA@IR-808, even at a high co-incubation concentration of 200 μ M. In contrast, the pure chromophores (IR-780 and IR-808) exhibited significant cytotoxicity at a co-incubation concentration of 30 μ M, further emphasizing the rationale and advantages of employing biomimetic FPs strategies to enhance dye biocompatibility.

Besides, we also included a discussion on the potential clinical applications of multicolor imaging in the revised manuscript. NIR imaging is the most clinically relevant fluorescence imaging technique due to its higher penetration depth and imaging contrast. Given the complexity and dynamic nature of organisms, single-color imaging alone is insufficient to fully capture the physiological and pathological changes. Real-time multicolor NIR-II imaging systems have the capability to address this limitation by simultaneously recording multiple events. This not only provides valuable insights into the functioning of organisms and the underlying mechanisms of diseases but also holds great promise in clinical practice for noninvasive diagnostics, image-guided surgery, and other applications. In summary, the implementation of multicolor imaging can effectively enhance bioimaging performance and imaging efficiency, with significant in vivo and clinical applicability.

[Redacted]

Figure R13. (a) NIR-II neuroimaging and (b) fluorescence signal statistics of the CO-1080 and HSA@CO-1080.

[Redacted]

Figure R14. (a) Imaging of nerve damage and (b) nerve fluorescence signal statistics of the HSA@CO-1080.

Figure R15. (a) Cytotoxicity of the IR-780 and $\beta\text{-LG@IR-780}$. (b) Cytotoxicity of the IR-808 and HSA@IR-808.

4. To validate the feasibility of the NIR-I/II FPs for multicolor bioimaging, the authors selected NIR-I protein-seeking dyes to construct biomimetic NIR-I FPs (HSA@IR-808 and $\beta\text{-LG@IR-780}$). What is the basis for selecting different protein shells? Please explain the binding mechanism of $\beta\text{-LG}$ with IR-780?

Response and revisions: Thank you for this valuable feedback. In this manuscript, we selected NIR-I protein-seeking dyes to construct biomimetic NIR-I FPs (HSA@IR-808 and $\beta\text{-LG@IR-780}$) and used in combination with NIR-II FPs to validate the feasibility of multi-color bioimaging. The choice of HSA and $\beta\text{-LG}$ proteins as the shell in this study was motivated by their ability to impart distinct pharmacokinetic behaviors to the probe, such as hepatobiliary or renal metabolism. Meanwhile, we successfully demonstrated the construction of the HSA@IR-808 and $\beta\text{-LG@IR-780}$ probes, showcasing the optimal pairing of different proteins with their respective dyes. The covalent binding between the proteins and dyes was highly efficient. This choice greatly contributed to verifying the feasibility of the biomimetic fluorescent proteins proposed in this work for multi-color

bioimaging.

Similar to the binding mechanism between HSA and CO-1080/IR-808, as shown in **Figure R16**, IR-780 was first inserted into the hydrophobic pocket of the protein by supramolecular interaction, followed by the covalent “clasp” through a nucleophilic substitution reaction between the Cl-C group of IR-780 and the -SH of Cysteine (Cys) on the β -LG protein. Meanwhile, we have further explored the specific binding sites of β -LG protein and IR-780 dye by proteomic analysis tests. As shown **Figure R17**, after careful analysis, the specific binding site between protein and dye was finally determined as Cys106.

Figure R16. Schematic of the bonding mechanism between β -LG and IR-780.

[Redacted]

Figure R17. Proteomic analysis of the β -LG@IR-780 probes.

5. In Figure 5, HSA@CO-1080 FPs demonstrated superior systemic and local (leg) vascular NIR-II imaging capabilities compared to the CO-1080 chromophores. The blood half-life of HSA@CO-1080 FPs should be described in this section, which is essential for angiography.

Response and revisions: Thanks for highlighting this important question. According to the reviewers' comments, we have added the blood half-life data for HSA@CO-1080 FPs in the revised manuscript. As shown in **Supplementary Fig. 39**, the HSA@CO-1080 FPs shows rapid metabolic behavior in the blood. Meanwhile, as shown in **Supplementary Fig. 40**, we have also characterized the time window for angiography of the HSA@CO-1080 FPs. Consistent with the results of blood half-life, the fluorescence of the corresponding vessels becomes negligible approximately 10 min after intravenous injection of HSA@CO-1080 FPs. We further investigated the repeating angiography ability of the HSA@CO-1080 FPs, as shown in **Supplementary Fig. 41**. As expected, HSA@CO-1080 FPs could achieve multiple/long-term monitoring of the blood vessels without significant interference from skin signals. These findings further confirmed the ability of HSA@CO-1080 FPs for long-term detection/tracing of vascular-related diseases, including skin grafts, arteriosclerosis, inflammatory vascular diseases, etc.

It should be noted that the overall excretion is difficult to estimate accurately due to the potential enzymatic degradation of the dye-albumin complex, as well as dissociation between the dye and albumin in vivo. Nevertheless, since albumin-derived formulations have been approved by the FDA for human use, we believe that our albumin-dye formulation also has the potential for clinical translation.

For angiography, the probe should have a relatively fast excretion/degradation rate post imaging. Non-specific binding of the probe can lead to high accumulation in organs and absorption through the skin, resulting in low-quality imaging and poor reproducibility of contrast during long-term monitoring. Therefore, HSA@CO-1080 is ideal for angiography as it allows for short-term repeatable injections for vascular imaging.

Supplementary Fig. 39. The blood half-life of HSA@CO-1080 FPs.

Supplementary Fig. 40. Time window for NIR-II angiography of the HSA@CO-1080 FPs.

Supplementary Fig. 41. (a) Repeat angiography images of the HSA@CO-1080 FPs. (b-d) Fluorescence signal statistics of repeat angiography for the HSA@CO-1080 FPs.

6. The encapsulation of NIR-I/II fluorophores with protein have been widely used as an effective approach to biomedical imaging and imaging-guided photothermal therapy (DOI: 10.1002/adhm.201800589). The obtained complexes also possess the improved biocompatibility and enhanced luminescence efficiency. In this manuscript, the authors fabricated the biomimetic near-infrared-II fluorescent proteins by covalently linking dyes CO-1080 with specific binding sites in proteins of HSA. This method lacks generalizability that could be generalized to other proteins and NIR-II chromophores. What is the advantage of biomimetic fluorescent proteins compared to the protein/dye complexes mentioned above?

Response and revisions: Thanks for highlighting this important question. Indeed, the encapsulation of NIR-I/II fluorophores with protein have been widely used as an effective approach to biomedical

imaging and imaging-guided photothermal therapy. Meanwhile, the protein encapsulation strategy has also been proved to significantly improve biocompatibility and enhance luminous efficiency of NIR-I/II fluorophores. However, it was worth noting that the biomimetic fluorescent proteins proposed in this work have the following advantages over the protein/dye complexes mentioned above:

(1) In contrast to traditional non-covalent encapsulation strategies, the biomimetic fluorescent protein proposed in this study utilizes a covalent binding approach between the protein and fluorophore. This innovative method ensures a high-efficiency covalent bond formation between the two components, effectively decreasing biosafety issues caused by dye leakage.

(2) Unlike previous protein/dye complexes that relied on physical encapsulation, the binding mechanism between the protein and dye molecules in our biomimetic fluorescent protein has been thoroughly investigated through multi-directional experiments. This approach allows for precise and controllable covalent binding behavior, resulting in a specific binding mechanism. Furthermore, the equimolar ratio of 1:1 between the protein and dye in the covalent binding strategy helps to avoid the ACQ phenomenon commonly observed in traditional dyes.

(3) The traditional protein encapsulation strategies often disrupted the original tertiary structure of the protein by using recombination between different peptide chains to encapsulate dye molecules. In contrast, the biomimetic construction strategy proposed in this work utilized a single protein molecule as the shell. By effectively utilizing the hydrophobic cavity inside the protein molecule and appropriate reaction sites, the strategy achieved one-to-one covalent entrapment of dye molecules. This approach maximized the retention of the intrinsic properties and biological functions of the protein itself.

(4) In our attempts to encapsulate CO-1080 dyes, we also explored the use of glutathione (GSH)-induced protein peptide chain recombination or ethanol-induced protein denaturation. Unfortunately, the brightness of the protein/dye complexes produced using these construction strategies was significantly lower compared to the biomimetic fluorescent proteins proposed in this study. Meanwhile, biomimetic fluorescent proteins could be synthesized under mild condition, according to the optimized results of reaction temperature (**Supplementary Fig. 6a-c**), it could be found that under the high temperature condition ($> 60^{\circ}\text{C}$), the fluorescence intensity of the formed protein/dye complexes decreased sharply due to the serious denaturation of the protein itself. Collectively, the biomimetic fluorescent protein construction strategy proposed in this manuscript effectively restricts the free torsion of dye molecules, resulting in a significant enhancement of fluorescence for CO-1080 dye molecules.

In addition, we acknowledge that the current biomimetic construction strategy may not be applicable to all proteins and NIR-II chromophores. However, we are actively working towards popularizing this strategy by screening proteins with high covalent gain effects, redesigning and synthesizing NIR-II chromophores, and further exploring the binding mechanism between proteins and dyes. To address the universality of the method, we have also conducted NHS modification of the 1080 series dye molecule. The amidation reaction is widely recognized as a more universal method for protein labeling. As shown in **Figure R18**, the protein/dye complexes bound by nucleophilic substitution reaction exhibited better luminescence ability compared to those bound by amidation reaction. This further confirms the advantages of the biomimetic construction strategy proposed in this work. In conclusion, we anticipate that through continuous exploration and research, this strategy will soon be widely promoted as a universal approach to protein tagging.

[Redacted]

Figure R18 (a) Reaction schematic of HSA and CE-1080 and CE-1080-NHS dyes. (b) NIR-II brightness of HSA and CE-1080 and CE-1080-NHS dyes after reaction at RT and 60°C for 2 h. (c) SDS-PAGE gel electrophoresis of HSA and CE-1080 and CE-1080-NHS dyes after reaction at RT and 60°C for 2 h.

7. What is the quantum yield of the CO-1080 dye and HSA@CO-1080 FPs?

Response and revisions: We are extremely grateful to the reviewer for pointing this issue out. As shown in **Supplementary Fig. 17** we have measured the quantum yield of the CO-1080 dye and HSA@CO-1080 FPs and added them in the revised manuscript. HSA@CO-1080 FPs have a relatively high QY compared with the free CO-1080 in DMSO.

Supplementary Fig. 17. UV-absorption spectroscopy of (a) IR-26 in dichloroethane (DCE), (b) CO-1080 in DMSO, and (c) HSA@CO-1080 at different gradient absorption. (d-e) Comparison of slope of IR-26 in dichloroethane (DCE), CO-1080 in DMSO, and HSA@CO-1080. (f) Quantum yields of CO-1080 in DMSO and HSA@CO-1080 (NIR-II, > 1100 nm).

8. What is the quality of in vivo fluorescence imaging with HSA@CO-1080 FPs? e.g., the depth of imaging, imaging resolution, signal-to-noise ratio?

Response and revisions: We are extremely grateful to the reviewer for pointing this issue out. We added in vivo fluorescence imaging quality data on the HSA@CO-1080 FPs in the revised manuscript, including imaging depth, imaging resolution, signal-to-noise ratio, and so on. As shown in **Figure 5b**, the HSA@CO-1080 FPs exhibited a signal-to-noise ratio as high as 42.71 when performing lymphatic imaging and was also able to clearly trace the lymphatic vessels. Moreover, the HSA@CO-1080 FPs also had some advantages in vascular imaging. As shown in **Figure 5d-f** and **Supplementary Fig. 35**, with the extension of the imaging window, its imaging depth, imaging resolution and signal-to-noise ratio have been greatly improved, with the maximum signal-to-noise ratio reaching 4.91 (> 1500 nm). Notably, the HSA@CO-1080 FPs demonstrated remarkable capabilities in fluorescence imaging. They exhibited negligible interference from skin/background signals under an imaging window larger than 1400 nm or 1500 nm. This allowed for clear visualization of renal contours deep in vivo, even at depths exceeding 5 mm. Besides, the HSA@CO-1080 FPs were also successfully applied to achieve high-precision vascular imaging in relatively large mammals such as rats and rabbits, which have thicker skin and adipose layers (**Supplementary Fig. 38**). Collectively, the HSA@CO-1080 FPs possessed high-quality NIR-II lymphography/ angiography capability in vivo, which was expected to accelerate the early clinical transformation of NIR-II fluorescence imaging technology.

Figure 5. (b) HSA@CO-1080 FPs-guided lymph node imaging and NIR-II-guided surgical excision (PLN: popliteal lymph node, SLN: sacral lymph node, > 1200 nm). (c) NIR-II lymphedema imaging of HSA@CO-1080 FPs (> 1200 nm). (d) NIR-II whole-body vessel imaging of HSA@CO-1080 FPs under different sub-NIR-II windows. (e-f) Cross-sectional fluorescence signals profile of the NIR-II whole-body vessels imaging.

Supplementary Fig. 38. (a) Angiographic schematic of different animals, including mouse, rat, and rabbit. (b) NIR-II angiography images of different animals after intravenous injection of the HSA@CO-1080 FPs, including mouse, rat, and rabbit. Statistics of blood vessel signal in (c) mouse, (d) rat, and (e) rabbit after intravenous injection of the HSA@CO-1080 FPs. All images were collected above 1200 nm (600 μ M).

Reviewer #3

Dear authors,

I have read carefully the manuscript submitted for consideration to NC.

The authors synthesized a new conjugate of cyanine dye and HSA. The topic is of interest. However, the conclusion is not supported by the data presented.

Response:

Thank you for your kind words and appreciation of our work. We value your suggestions for further improving our research. Below, we provide a detailed response to each of your comments.

1. The binding with protein at high temperature can affect the binding ability. Have you performed the experiments dye-protein noncovalent binding, like measuring binding constants with native HSA and heated?

Response: Thanks for highlighting this important question. In fact, as shown in **Supplementary Fig. 6a-c**, temperature had a significant effect on the binding ability between dyes and proteins. It could be seen from the SDS-PAGE electrophoresis data that the HSA@CO-1080 complex probe showed a negligible fluorescence band at the corresponding molecular weight position under the reaction condition of 30°C or RT, which also indicated that there was no effective covalent bond between protein and dyes. In other words, a non-covalent binding occurred between the protein and dye at 30°C or RT. Meanwhile, with the increase of reaction temperature, the covalent binding effect of HSA@CO-1080 complex was significantly improved. And at the reaction condition of 60°C, the brightest fluorescence band was displayed at the corresponding molecular weight position, showing the most outstanding covalent binding effect. With the further increase of reaction temperature, the covalent bond effect between protein and dyes decreased greatly because of the denaturation of protein. Meanwhile, as shown in **Figure R19**, NIR-II brightness and SDS-PAGE gel electrophoresis consistently showed that HSA and different dyes could be directly mixed at room temperature to form the non-covalent complexes. It was worth emphasizing that the NIR-II fluorescence intensity of different protein-dye non-covalent complexes was much lower than that of their corresponding covalent complexes and exhibited negligible fluorescence bands at the corresponding molecular weight positions.

We also compared the UV absorption and fluorescence of dyes non-covalent/covalent binding with protein. As shown in **Figure R20**, compared with the non-covalent binding method, the covalent binding method could effectively avoid the H or J aggregation of pure dye molecules in an aqueous solution and significantly improved their NIR-II emission ability. Besides, as shown in **Figure 4g, h** and **Supplementary Fig. 29**, the non-covalent (glide docking) and covalent binding (covalent docking) behaviors between HSA and CO-1080 were also compared by theoretical simulation data. Results showed that the binding energy between protein and dye by covalent binding mode was higher than that by non-covalent binding mode, which made the covalent complex show more prominent luminescence ability.

In addition, we characterized the non-covalent/covalent binding of CO-1080 with native HSA (RT) and heated HSA (60°C and 90°C) by the high-resolution mass spectrometry. As shown in **Figure R21**, consistent with the above SDS-PAGE electrophoresis results, the covalent bond between CO-1080 and native HSA was negligible, indicating a non-covalent interaction between the dye and the protein. In contrast, CO-1080 exhibited a high-efficient covalent bond with HSA heated to 60°C. However, at higher heating conditions (90°C), the sample purity was limited due to

the protein denaturation and thermal instability of dye molecules, making it impossible to provided high-resolution mass spectrometry data with high confidence.

Supplementary Fig. 6. (a) NIR-II fluorescence intensity, (b) SDS-PAGE electrophoresis analysis, and (c) signal statistics of HSA@CO-1080 under different reaction temperatures.

Figure R19. (a) NIR-II brightness and (b) SDS-PAGE gel electrophoresis after non-covalent (mix directly at RT) and covalent binding (60°C, 2 h) between HSA and Et-1080, CO-1080, and FD-1080. (NC: non-covalent binding, C: covalent binding)

Figure R20. UV absorption spectra of HSA after (a) non-covalent binding and (b) covalent binding with different dyes, including Et-1080, CO-1080, and FD-1080. Fluorescence spectroscopy of HSA after (c) non-covalent binding and (d) covalent binding with different dyes, including Et-1080, CO-1080, and FD-1080.

Figure R21. High-resolution mass spectrometry of (a) HSA, (b) HSA@CO-1080 (RT, 2 h), (c) HSA@CO-1080 (60°C, 2 h), and (d) HSA@CO-1080 (90°C, 2 h).

Note: For the HSA@CO-1080 (90°C, 2 h), the high-resolution mass spectrometry data with high reliability could not be provided due to unclear baseline of the original spectra, low signal-to-noise ratio, and lack of baseline separation between individual peaks.

2. a. Please provide excitation and emission spectra of dyes and their complexes with protein, both covalent and non-covalent complexes.

Response and revisions: Thank you for this valuable suggestion. According to the comments of the reviewer, we have added the emission spectra of dyes and their complexes with protein, including covalent and non-covalent complexes, in the revised manuscript (**Supplementary Fig.13**

and Supplementary Fig.14). Especially, for the excitation spectra, due to the lack of external excitation module in the configuration of the instrument, the excitation spectra of different samples could not be provided here. We sincerely apologize and hope that the reviewers can understand. Nevertheless, we have conducted experiments to measure the brightness of the sample under different excitation wavelengths, which can indirectly reflect the optimal excitation wavelength for the sample.

Supplementary Fig. 13. (a) Photographs of HSA@Et-1080, HSA@CO-1080, and HSA@FD-1080 before and after the reaction. (b) UV-absorption spectroscopy and (e) fluorescence spectroscopy of the HSA and Et-1080 before and after reaction. (c) UV-absorption spectroscopy and (f) fluorescence spectroscopy of the HSA and CO-1080 before and after reaction. (d) UV-absorption spectroscopy and (g) fluorescence spectroscopy of the HSA and Et-1080 before and after reaction.

Note: HSA and different dyes before reaction (RT, direct mixing) are also defined as non-covalent binding. HSA and different dyes after reaction (60°C, 2 h) are also defined as covalent binding.

Supplementary Fig. 14. (a) UV-absorption spectroscopy and (d) fluorescence spectroscopy of Et-1080 in DMSO solution, Et-1080 in PBS solution, and HSA@Et-1080. (b) UV-absorption spectroscopy and (e) fluorescence spectroscopy of CO-1080 in DMSO solution, CO-1080 in PBS solution, and HSA@CO-1080. (c) UV-absorption spectroscopy and (f) fluorescence spectroscopy of FD-1080 in DMSO solution, FD-1080 in PBS solution, and HSA@FD-1080.

2. b. The J-aggregate formation is also possible in regarded conditions. It is necessary to verify that observed fluorescence is not emitted by aggregates.

Response: Thanks for highlighting this important question. Indeed, we successfully discovered that the CO-1080 dyes could form j-aggregates under specific conditions. As shown in **Supplementary Fig. 15.**, the UV absorption curves of CO-1080 dyes under different conditions clearly demonstrate that the dyes exhibit H-aggregation behavior in PBS solution without any reaction treatment. However, after being heated at 60°C for 2 hours in PBS solution, the CO-1080 dyes surprisingly exhibit J-aggregation behavior. Conversely, in HSA solution, the CO-1080 dyes display a monodisperse behavior similar to that in DMSO solution after being heated at 60°C for 2 hours. Meanwhile, by comparing the fluorescence intensity of CO-1080 dyes under the aforementioned three reaction conditions, it could be also clearly found that the CO-1080 dyes only showed the bright NIR-II luminescence after being heated at 60°C for 2 h in HSA solution (**Figure R22**). In contrast, they showed negligible luminescence ability under the other two reaction conditions (H-aggregation and J-aggregation). Collectively, it could be effectively confirmed that the fluorescence observed by imaging with the protein@dye complex proposed in this work was not emitted by aggregates.

Supplementary Fig. 15. Schematic of the structure and properties of CO-1080 dyes under different reaction conditions.

Figure R22. Comparison of fluorescence intensity of CO-1080 dyes under different reaction conditions.

3. According to proposed binding mechanism, the dyes have identical chromophore structure. Thus, it is unobvious the CO-1080 binds to HSA most effective, because of steric hindrance.

Response: Thanks for highlighting this important question. As shown in **Figure R23**, the ET-1080, CO-1080, and FD-1080 dyes have similar chromophore structure, but different side chains. However, a careful comparison revealed that the difference in side chains had a significant impact on the NIR-II luminescence ability of the different dye molecules. In particular, the CO-1080 dyes exhibited a more prominent NIR-II brightness (**Figure 2b** and **Supplementary Fig. 12**). Notably, as shown in **Figure 2c-I**, the differences in side chains directly affected the luminescence and covalent binding ability of the protein. The HSA@CO-1080 exhibited a more pronounced NIR-II brightness, approximately 3.51-fold that of HSA@Et-1080 and 2.72-fold that of HSA@FD-1080. This finding was further supported by SDS-PAGE gel electrophoresis and high-resolution mass spectrometry results, which consistently confirmed that HSA@CO-1080 had the highest covalent binding efficiency of 88.98%, while Et-1080 and FD-1080 dyes showed suboptimal binding efficiencies of 51.84% and 5.31% with HSA (**Fig. 2k, l** and **Supplementary Fig. 19**). Besides, the molecular docking simulation results using the glide program further supported the notion that

different side chains have a direct impact on the binding behavior between HSA and the Et-1080, CO-1080, and FD-1080 dyes. The results showed that CO-1080 had the largest number of amino acid interactions, as well as the highest docking score and binding energy. This indicates that CO-1080 has a stronger binding affinity for HSA compared to Et-1080 and FD-1080 dyes (**Fig. 2f-j**).

We also compared in detail the effects of different side chains on the binding behavior of 1080 series dyes and proteins. As shown in **Figure R6**, our findings revealed that as the side chain length increased, the NIR-II brightness and covalent bond efficiency of the HSA@CO-1080-C_n dye complexes initially increased and then decreased. Among them, CO-1080-C₆ (CO-1080) showed the best binding behavior with HSA. Meanwhile, when comparing the 1080 series dyes with different side chain end groups, we observed that the end groups significantly affected the binding behavior between proteins and dyes (**Figure R7**). The binding behavior between 1080 series dyes with carboxyl end group (CO-1080) and protein was obviously better than that of 1080 series dyes with alkylend group (Et-1080-C₆). Collectively, the differences in binding behavior between the different 1080 series dyes (Et-1080, CO-1080, and FD-1080) and proteins were not primarily caused by steric hindrance, but rather were mainly influenced by the differences in side chains, including side chain length and side chain end groups.

Figure R23. Molecular structure of the Et-1080, CO-1080, and FD-1080 dyes.

Figure R6. (a) Reaction schematic of the HSA and CO-1080 dyes with different side chain lengths. (b) NIR-II brightness of the CO-1080-Cn in PBS and HSA@CO-1080-Cn. (c) SDS-PAGE gel electrophoresis of the CO-1080-Cn in PBS and HSA@CO-1080-Cn.

Figure R7. (a) Molecular structure of the Et-1080-C6 and CO-1080 dyes. (b) NIR-II brightness of Et-1080-C6 and CO-1080 in PBS and HSA. (c) SDS-PAGE gel electrophoresis of HSA@Et-1080-C6 and HSA@CO-1080.

4. Comparison of dyes non-covalent binding with protein and binding constants need to be provided.

Response: Thank you for this valuable suggestion. According to the comment, we have added the comparison of dyes non-covalent binding with protein and binding constants. We first performed the Bio-Layer Interferometry (BLI) to evaluate the binding affinity between the HSA and 1080 series dyes. As shown in **Supplementary Fig. 18** and **Supplementary Table 2**, the CO-1080 exhibited the most prominent binding affinity to HSA with K_D of ~ 2.13 nM, which was far superior to Et-1080 (14.5 nM) and FD-1080 (82.6 nM).

In addition, the non-covalent binding and covalent binding of different dyes to proteins were investigated in detail. As shown in **Figure R19, 24**, and **Supplementary Fig. 19**, NIR-II brightness, SDS-PAGE gel electrophoresis, and high-resolution mass spectrometry consistently showed that HSA and different dyes could be directly mixed at room temperature to form the non-covalent complexes. It was worth emphasizing that, for different protein-dye non-covalent complexes, the HSA@CO-1080 non-covalent complexes still exhibited relatively superior fluorescence intensities, consistent with the trend for covalent complexes. Furthermore, compared with the non-covalent complexes, the NIR-II fluorescence intensity of the covalent complexes was greatly enhanced. Besides, as shown in **Figure 2f-j**, the gliding docking simulation data also reflected the non-covalent binding behavior between HSA and different dyes. Results revealed that CO-1080 had the largest number of amino acid interactions, as well as the highest binding score and binding energy, surpassing the binding of Et-1080 and FD-1080 dyes. These non-covalent simulation findings strongly implied that HSA exhibited stronger free-twisting restriction on the CO-1080 dyes compared to the Et-1080 and FD-1080 dyes. This led to a more effective reduction in energy dissipation caused by non-radiative transitions, thus resulting in the optimal NIR-II brightness of the HSA@CO-1080 complex regardless of whether it was non-covalent or covalent bound. Additionally, we also compared the UV absorption and fluorescence of dyes non-covalent/covalent binding with protein. As shown in **Figure R20**, whether non-covalently or covalently bonded HSA@CO-1080 showed the most significant improvement in absorbance and NIR-II fluorescence emission. Moreover, compared with the non-covalent binding method, the covalent binding method could effectively avoid the H or J aggregation of pure dye molecules in an aqueous solution and significantly improved their NIR-II emission ability.

Supplementary Fig. 17. Kinetic binding assay of (a) Et-1080, (b) CO-1080, and (c) FD-1080 with HSA using bio-layer interferometry technique.

Supplementary Table 2. K_D , K_{on} and K_{off} for Et-1080, CO-1080, and FD-1080 with HSA

Protein	Dye	K_D (nM)	K_{on}	K_{off}
HSA	Et-1080	14.5	1.04E+03	1.52E-02
	CO-1080	2.13	1.09E+04	2.32E-02
	FD-1080	82.6	3.04E+03	2.51E-01

K_D : binding affinity; K_{on} : binding constant; K_{off} : dissociation constant

Figure R19. (a) NIR-II brightness and (b) SDS-PAGE gel electrophoresis after non-covalent (mix directly at RT) and covalent binding (60°C, 2 h) between HSA and Et-1080, CO-1080, and FD-1080. (NC: non-covalent binding, C: covalent binding)

Figure R24. (a) High-resolution mass spectrometry of HSA mixed directly with Et-1080, CO-1080 and FD-1080 at room temperature.

Figure R20. UV absorption spectra of HSA after (a) non-covalent binding and (b) covalent binding with different dyes, including Et-1080, CO-1080, and FD-1080. Fluorescence spectroscopy of HSA after (c) non-covalent binding and (d) covalent binding with different dyes, including Et-1080, CO-1080, and FD-1080.

5. The fluorescence lifetime for each dye and both types of complexes can significantly improve manuscript.

Response: Thank you for this valuable suggestion. We attempted to measure the fluorescence lifetime of the dyes and their complexes using various available facilities. Unfortunately, as shown in **Figure R25**, due to the unique characteristics of the dyes' maximum emission wavelength 1000 nm and the limitations of the instruments used for fluorescence lifetime measurements (with a maximum excitation wavelength of 655 nm), we were unable to detect any significant fluorescence signal. Consequently, the fluorescence lifetime of both the pure dyes and their corresponding complexes could not be determined.

Figure R25. The fluorescence lifetime for CO-1080 in DMSO, CO-1080 in PBS, and HSA@CO-1080.

6. The site-specific binding with FBZ and Ibuprofen can confirm the localization in DIII site. It can support the docking information.

Response: Thank you for this valuable suggestion. Indeed, previous reports have successfully demonstrated that the specific binding sites of FBZ and ibuprofen to HSA were DIII domains. In many studies, researchers have also used FBZ and/or ibuprofen to indirectly confirm the protein-ligand interaction site (e.g., *small* **2022, 18, 2201298**). Instead, in this work, we directly used three mutated domain proteins (DI, DII, and DIII) of HSA for dye-binding site verification. As shown in **Figure 3**, the fluorescence enhancement, SDS-PAGE gel electrophoresis, and high-resolution mass spectrometry data consistently confirmed that the major binding site of 1080 series dyes and HSA was DIII. The above conclusion also fully supported the docking simulation information.

In addition, we attempted to use ibuprofen (IBU) to confirm the binding site between HSA and CO-1080. The experimental procedure was as follows: IBU was selected as the preferred compound to mix with HSA. After incubating at room temperature for 2 h, the CO-1080 dyes were added to the above mixed solution for a covalent binding reaction (**Figure R26**). To be sure, as shown in **Figure R27**, the presence of IBU did affect the binding of HSA and CO-1080 dye, especially when IBU was in excess with a molar ratio of IBU to HSA of 5:1. However, due to the non-covalent interaction between IBU and HSA, the presence of IBU did not significantly hinder the highly efficient covalent binding between HSA proteins and CO-1080 dyes. Therefore, instead of using FBZ or ibuprofen to confirm binding sites, it would be more accurate and convincing to directly utilize different domain (DI, DII, and DIII) proteins of HSA to confirm the specific binding sites between proteins and dyes.

Figure R26. Experimental flow chart between IBU, CO-1080, and HSA.

Figure R27. (a) Fluorescence intensity after reaction of HSA protein and CO-1080 in the presence of different IBU content. (b) SDS-PAGE gel electrophoresis after reaction of HSA protein and CO-1080 in the presence of different IBU content.

7. The absence of photodynamic activity need to be proven. The triplet state formation and singlet oxygen generation in nonpolar organic solvents and PBS with Dye-HSA complex. Need to be compared with using CO-1080 separately.

Response: Thank you for this valuable suggestion. Based on the comments of the reviewer, we investigated the photodynamic activities of CO-1080 and HSA@CO-1080 probes. **Figure R28** showed the results of our assessment of the potency of the CO-1080 and HSA@CO-1080 probes in terms of the generation upon 1064 nm laser irradiation, using a dichlorodihydrofluorescein (DCFH) probe. The presence of the CO-1080/HSA@CO-1080 probe led to a significant enhancement in the fluorescence intensity of DCFH at 525 nm with increasing irradiation time, confirming the successful generation of ROS. Furthermore, through careful comparative analysis, we observed that HSA@CO-1080 exhibited superior ROS-generating capacity and demonstrated more prominent photodynamic activity compared to pure CO-1080 dyes.

Figure R28. Reactive oxygen species (ROS) generation by the (a) CO-1080 in PBS and (b) HSA@CO-1080 (DCFH as a probe, 488 nm excitation).

To get a better insight into the nature of the generated ROS, singlet oxygen (1O_2), superoxide radical ($O_2^{\cdot-}$), and hydroxyl radical ($\cdot OH$) generations were respectively detected with 1,3-diphenylisobenzofuran (DPBF) probe, dihydrorhodamine 123 (DHR 123) probe, and hydroxyphenyl fluorescein (HPF) probe. As shown in **Figure R29**, whether pure CO-1080 or HSA@CO-1080, the ROS generation was mainly related to singlet oxygen (1O_2) and superoxide radicals ($O_2^{\cdot-}$), and was dominated by superoxide radicals ($O_2^{\cdot-}$). Furthermore, it was emphasized that the photodynamic activity of Dye-HSA complexes in non-polar organic solvents is limited due to the intolerance of proteins to such solvents. However, in our study, we compared the

photodynamic activity of CO-1080 dyes in different solutions, including polar organic solvent (DMSO), nonpolar organic solvent (dichloroethane, DCE), PBS, and HSA. The results clearly demonstrated that the order of photodynamic activity of CO-1080 dyes in these different solutions was as follows: HSA > PBS > DCE > DMSO. This indicates that the presence of HSA significantly enhances the photodynamic activity of CO-1080 dyes, and the photodynamic activity is influenced by the nature of the solvent used.

Figure R29. (a) Singlet oxygen ($^1\text{O}_2$) generation with the presence of the CO-1080 in DMSO (polar solvent), CO-1080 in DCE (nonpolar solvents), CO-1080 in PBS, and HSA@CO-1080 (DPBF as a probe, 488 nm excitation). (b) Superoxide radical ($\text{O}_2^{\cdot-}$) generation with the presence of the CO-1080 in DMSO (polar solvent), CO-1080 in DCE (nonpolar solvents), CO-1080 in PBS, and HSA@CO-1080 (DHR 123 as a probe, 488 nm excitation). (c) Hydroxyl radical ($\cdot\text{OH}$) generation with the presence of the CO-1080 in DMSO (polar solvent), CO-1080 in DCE (nonpolar solvents), CO-1080 in PBS, and HSA@CO-1080 (HPF as a probe, 488 nm excitation).

Minor:

line 60 and 61 chromophobe need to be replaced with chromophore

Response and revisions: We are extremely grateful to the reviewer for pointing this issue out. We have carefully re-checked the manuscript for corresponding errors and made corrections.

What means PL intensity on graph? Fluorescence?

Response and revisions: We are extremely grateful to the reviewer for pointing this issue out. Indeed, PL intensity is an abbreviation for photoluminescence intensity, which is also synonymous with fluorescence intensity for dye molecules. In order to eliminate any confusing, we have thoroughly reviewed the manuscript and replaced all instances of "PL intensity" with "FL intensity".

Reviewers' Comments:

Reviewer #1:

None

Reviewer #2:

Remarks to the Author:

This study reports a chemogenic protein-seeking strategy for fabricating biomimetic near-infrared-II fluorescent proteins by covalently linking dyes and specific binding sites in proteins. The obtained HSA@CO-1080 FPs enabled integrated two-color NIR-II imaging of various biological events without interference, such as lymph node colocalization, lymphatic-vascular visualization, and pharmacokinetics assessment. Most of the concerns that we proposed have been properly addressed. However, some issues need to be addressed before it is accepted for publication:

1. The legends in Figures S16 and S17 are wrong. In Figure S16, the absorption number of f) HSA@Et-1080 and h) HSA@FD-1080 is the same. In Figure S17, the absorption number of b) CO-1080 in DMSO and c) HSA@CO-1080 is the same. Please revise this part.

2. Question 5, the authors claimed that "HSA@CO-1080 is ideal for angiography as it allows for short-term repeatable injections for vascular imaging". However, according to the results, the fluorescence of the corresponding vessels becomes negligible approximately 10 min after intravenous injection of HSA@CO-1080 FPs, which is quite short for angiography, especially for clinical applications. In addition, what is the biosafety of short-term repeatable injections with HSA@CO-1080 FPs? Although HSA@CO-1080 FPs showed better biocompatibility compared to free dyes, the potential biotoxicity of short-term repeatable injections of HSA@CO-1080 FPs increases owing to the increased dosage. In addition, *in vivo* degradation also resulted in the release of some free dye, which may further increase the potential biotoxicity.

3. Question 7, the authors claimed that HSA@CO-1080 FPs have a relatively high QY compared with the free CO-1080 in DMSO. However, the data in Supplementary Fig. 17 showed the opposite results, the quantum yield (>1100 nm) of free CO-1080 in DMSO (0.2664%) is higher than that of HSA@CO-1080 (0.0921%). How do the authors calculate the quantum yield?

4. Question 8, the authors claimed that they added the *in vivo* fluorescence imaging quality data on the HSA@CO-1080 FPs in the revised manuscript, including imaging depth. However, I did not find the related data. The only data I found related to imaging depth are shown in Supplementary Fig. 9 and Supplementary Fig. 10. But these works were processed *in vitro* not *in vivo*. Again, the authors claimed that "This allowed for clear visualization of renal contours deep *in vivo*, even at depths exceeding 5 mm." I did not find any relevant data or figures. How did the authors conclude that "in *in vivo* imaging depths exceeding 5 mm"? The authors should point out the detailed revision section in the manuscripts or Supplementary materials. In some sections, it seems that the authors only responded to the reviews' questions, but did not make revisions to their articles.

Reviewer #3:

Remarks to the Author:

The outstanding amount of provided experimental data is beneficial for manuscript strength. The chemical binding of NIR dyes to albumin can become very useful technique for medicine. Authors have significantly improved the quality of manuscript by providing additional data. A few moments described below are crucial for understanding the mechanism of dye-protein interaction:

1. In figure r20 was shown the difference between the main absorbance band for covalent-binded dye and the non-covalent binding. How the observed bathochromic shift can be explained?

2. The demonstrated spectra at figure 13 indicates that the main difference between dyes is solubility in water and aggregate formation. Does covalent binding efficiency correlates with amount of dye in monomeric form?

3. The observed value of dye fluorescence lifetime is very short which is common for aggregates. The increase in fluorescence intensity from 40k (noncovalent) to 200k(covalent) for CO-1080 with

no significant change in fluorescence lifetime for equal amount of compound. Is it possible to add to non-covalent and covalent binding with HSA some amount of tween-80 or related reagent to confirm mechanism of disaggregation, provided in article?

4. The value of measured singlet oxygen generation in solutions of protein without dye like blank standard samples and in nonpolar aprotic solvent like toluene can significantly improve the quality of manuscript.

Point by point replies to reviewers and revisions made.

Reviewer #2 (Remarks to the Author):

This study reports a chemogenic protein-seeking strategy for fabricating biomimetic near-infrared-II fluorescent proteins by covalently linking dyes and specific binding sites in proteins. The obtained HSA@CO-1080 FPs enabled integrated two-color NIR-II imaging of various biological events without interference, such as lymph node colocalization, lymphatic-vascular visualization, and pharmacokinetics assessment. Most of the concerns that we proposed have been properly addressed. However, some issues need to be addressed before it is accepted for publication:

Response:

We thank the reviewers for endorsing our last round of revisions. We also highly appreciate the reviewer's suggestions for further strengthening our work. Here, we listed a detailed response to the reviewer's comments.

1. The legends in Figures S16 and S17 are wrong. In Figure S16, the absorption number of f) HSA@Et-1080 and h) HSA@FD-1080 is the same. In Figure S17, the absorption number of b) CO-1080 in DMSO and c) HSA@CO-1080 is the same. Please revise this part.

Response and revisions: We sincerely appreciate the reviewer for bringing this issue to our attention. We deeply regret making such a basic mistake. We have thoroughly reviewed the manuscript and have made the necessary corrections to address this kind of issue.

2. Question 5, the authors claimed that “HSA@CO-1080 is ideal for angiography as it allows for short-term repeatable injections for vascular imaging”. However, according to the results, the fluorescence of the corresponding vessels becomes negligible approximately 10 min after intravenous injection of HSA@CO-1080 FPs, which is quite short for angiography, especially for clinical applications. In addition, what is the biosafety of short-term repeatable injections with HSA@CO-1080 FPs? Although HSA@CO-1080 FPs showed better biocompatibility compared to free dyes, the potential biotoxicity of short-term repeatable injections of HSA@CO-1080 FPs increases owing to the increased dosage. In addition, in vivo degradation also resulted in the release of some free dye, which may further increase the potential biotoxicity.

Response: Thank you for bringing up this important question. We acknowledge that the angiographic time window for HSA@CO-1080 FPs was relatively short, with negligible fluorescence of the corresponding vessels approximately 10 minutes after intravenous injection. Notably, this characteristic allows for short-term repeat-injection angiography, which is beneficial for one-stop preoperative protocol development, intraoperative navigation, and postoperative assessment of blood path recovery. We would like to emphasize that our proposed biomimetic fluorescent protein construction strategy is indeed feasible, and the angiographic time window for different biomimetic fluorescent proteins can be adjusted based on the protein coating or chromophore properties. For instance, as shown in **Figure R1**, the as-prepared BSA@IR-780 exhibited a prominent angiographic time window of up to 3 hours, while β -LG@IR-780 and HSA@CO-1080 demonstrated similar angiographic behavior with a relatively short time window.

Therefore, we are continuously working on developing more biomimetic fluorescent proteins with different performance advantages. This allows us to select the most suitable developer for imaging applications based on specific disease requirements.

Figure R1. The angiographic time window of (a) BSA@IR-780 and (a) β -LG@IR-780.

Meanwhile, we have also conducted a biosafety evaluation of short-term repeatable injections with HSA@CO-1080 FPs. **Figure R2** presents the results of various biochemical analyses, including blood routine, hepatic function, and kidney function. These results further confirm the outstanding biological safety of HSA@CO-1080 FPs, as even after short-term multiple injections, no significant injury to the body was observed.

Figure R2. (a) Flow chart of the HSA@CO-1080 FPs for short-term repeated injections. (b) Blood routine indexes after short-term repeated injection of the HSA@CO-1080 FPs. (c) Hepatic and (d) renal function indexes after short-term repeated injection of the HSA@CO-1080 FPs

3. Question 7, the authors claimed that HSA@CO-1080 FPs have a relatively high QY compared with the free CO-1080 in DMSO. However, the data in Supplementary Fig. 17 showed the opposite results, the quantum yield (>1100 nm) of free CO-1080 in DMSO (0.2664%) is higher than that of HSA@CO-1080 (0.0921%). How do the authors calculate the quantum yield?

Response and revisions: Thank you for this valuable feedback. To clarify, for question 7, the confusion arises from our misrepresentation. In fact, we would like to state that the quantum yield

(> 1100 nm) of HSA@CO-1080 FPs (0.0921%) does not differ significantly from that of free CO-1080 in DMSO (0.2664%), maintaining an approximate value of 34.57%. Meanwhile, we have included a detailed calculation steps for the quantum yields of CO-1080 and HSA@CO-1080 FP in the revised manuscript, as follows:

The quantum yields of free CO-1080 in DMSO and HSA@CO-1080 FPs were determined using the dye IR-26 ($\Phi_f = 0.5\%$ in 1,2-dichloroethane (DCE)) as reference. Briefly, a gradient dilution was performed for IR-26, CO-1080, and HSA@CO-1080 in different solvents, ensuring that the OD at 1064 nm was less than 0.1. The corresponding fluorescence intensity was measured under excitation at 1064 nm (> 1100 nm). The integrated fluorescence intensity was plotted as a function of the OD value at 1064 nm and fitted into a linear function. The slopes of the linear fit for IR-26 in DCE, CO-1080 in DMSO, and HSA@CO-1080 FPs in PBS were obtained. The quantum yields of CO-1080 in DMSO and HSA@CO-1080 FPs in PBS were calculated using the following equation:

$$QY_{\text{sample}} = QY_{\text{ref}} \times \left(\frac{\eta_{\text{sample}}}{\eta_{\text{ref}}}\right)^2 \times \frac{\text{Slope}_{\text{sample}}}{\text{Slope}_{\text{ref}}}$$

where QY_{sample} is the quantum yield of CO-1080 in DMSO or HSA@CO-1080 FPs in PBS, QY_{ref} is the quantum yield of IR-26 in DCE, η_{ref} and η_{sample} are the refractive indices of corresponding solvents. $\text{Slope}_{\text{sample}}$ and $\text{Slope}_{\text{ref}}$ are the slopes obtained by linear fitting of the integrated fluorescence intensity of IR-26 in DCE, CO-1080 in DMSO, and HSA@CO-1080 FPs in PBS.

4. Question 8, the authors claimed that they added the in vivo fluorescence imaging quality data on the HSA@CO-1080 FPs in the revised manuscript, including imaging depth. However, I did not find the related data. The only data I found related to imaging depth are shown in Supplementary Fig. 9 and Supplementary Fig. 10. But these works were processed in vitro not in vivo. Again, the authors claimed that “This allowed for clear visualization of renal contours deep in vivo, even at depths exceeding 5 mm.” I did not find any relevant data or figures. How did the authors conclude that “in vivo imaging depths exceeding 5 mm”? The authors should point out the detailed revision section in the manuscripts or Supplementary materials. In some sections, it seems that the authors only responded to the reviews’ questions, but did not make revisions to their articles.

Response: Thank you for this valuable feedback. Indeed, apart from the in vitro imaging depth data in **Supplementary Fig. 9** and **Supplementary Fig. 10**, there were several in vivo data related to imaging depth of the HSA@CO-1080 FPs in the manuscript. As shown in **Figure 5b**, the HSA@CO-1080 FPS allowed for precise localization and visualization of deep subcutaneous lymph nodes.

Figure 5b. HSA@CO-1080 FPs-guided lymph node imaging and NIR-II-guided surgical excision.

I apologize for the oversight in my previous response. As shown in **Figure 5d** and **Figure R3**, when performing systemic vascular imaging with an imaging window larger than 1400 nm, it was able to clearly visualize not only the blood vessels on the body surface but also penetrate the epidermis, revealing the contour of deep-seated renal structures. This provides strong evidence of the imaging depth capabilities of the HSA@CO-1080 FPs in vivo.

Figure R3. NIR-II whole-body vessel imaging of HSA@CO-1080 FPs under the imaging window larger than 1500 nm.

As shown in **Supplementary Fig. 38**, the HSA@CO-1080 FPs were also successfully applied to achieve high-precision vascular imaging in relatively large mammals such as rats and rabbits, which have thicker skin and adipose layers. The above results also effectively confirmed the ability of the HSA@CO-1080 FPs in deep imaging.

Supplementary Fig. 38. (a) Angiographic schematic of different animals, including mouse, rat, and rabbit. (b) NIR-II angiography images of different animals after intravenous injection of the HSA@CO-1080 FPs, including mouse, rat, and rabbit. Statistics of blood vessel signal in (c) mouse, (d) rat, and (e) rabbit after intravenous injection of the HSA@CO-1080 FPs. All images were collected above 1200 nm (600 μ M).

In addition, the conclusion that the depth of imaging in vivo exceeds 5 mm was based on the observation that the kidney was more than 5 mm away from the body surface during dissection in mice (**Figure R4**). In order to further illustrate the deep imaging capability of ability of the HSA@CO-1080 FPs in vivo, another set of tissue penetration experiment was carried out by selecting chicken breast as a living substitute. As shown in **Figure R5**, the HSA@CO-1080 FPs still possessed excellent imaging ability even at a thickness of over 7 mm.

We would also like to clarify that not all of our responses to the reviewer's comments were included in the manuscript. Due to limitations in article length and data relevance, we only incorporated key data set into the revised manuscript. However, it is important to note that the journal follows a transparent peer review process, and all our responses to the reviewers' comments will be published as a supplementary peer review file along with the article.

Figure R4. Image of mouse kidney depth.

Figure R5. Comparison of the tissue penetration depth of the HSA@CO-1080 under (a) 0 mm, (b) 1 mm, (c) 3 mm, (d) 5 mm, (e) 7 mm, and 10 mm chicken breast thickness.

Reviewer #3 (Remarks to the Author):

The outstanding amount of provided experimental data is beneficial for manuscript strength. The chemical binding of NIR dyes to albumin can become very useful technique for medicine. Authors have significantly improved the quality of manuscript by providing additional data. A few moments described below are crucial for understanding the mechanism of dye-protein interaction:

Response:

We would like to express our gratitude for the positive feedback on our previous revisions. We also highly value the reviewer's suggestions for further enhancing our work. In response to the reviewer's comments, we have provided a detailed response below.

1. In figure r20 was shown the difference between the main absorbance band for covalent-binded dye and the non-covalent binding. How the observed bathochromic shift can be explained?

Response: Thank you for this valuable feedback. Indeed, the difference between the main absorbance band for the covalently bound and non-covalently bound dyes was due to the self-aggregation behavior of the dyes and the interaction between proteins and dyes. As shown in **Supplementary Figs 14, 15**, due to the special structure of dye molecules, self-aggregation often occurred in aqueous solution, resulting in red shift (J-aggregation) and blue shift (H-aggregation) of absorption. In the presence of protein, the aggregation behavior of pure dyes could be effectively suppressed, resulting in dye absorption changed to varying degrees. For the non-covalent binding between a protein and a dye, the aggregation behavior often relies on the affinity between the dye and the protein. The stronger the binding between the two, the closer the maximum absorption peak of the protein-dye complex will be to the absorption value of the dye molecule in the organic phase. In the case of covalent binding between a protein and a dye, the dye molecules are effectively trapped within the protein cavity, leading to the complete disbandment of their aggregation behavior and the formation of a monodisperse state. At this stage, the peak shapes of protein-dye complexes and pure dye molecules in the organic phase are almost identical. Therefore, different dye molecules observed different bathochromic shifts under different binding conditions with HSA.

2. The demonstrated spectra at figure 13 indicates that the main difference between dyes is solubility in water and aggregate formation. Does covalent binding efficiency correlate with amount of dye in monomeric form?

Response: Thanks for highlighting this important question. According to the reviewers' comments, we investigated the relationship between the covalent binding efficiency and the number of dyes with monomeric form. As shown in **Figure R6a**, CO-1080 in DMSO as the stock solution of dye monomers and CO-1080 in PBS was selected as the stock solution of dye aggregates. Then the above different stock solutions were reacted respectively with HSA to investigate the covalent binding differences between them. Results showed that the covalent binding efficiency and luminescence between HSA and dyes were affected by the aggregation behavior. The results indicate

that as the dye molecule approaches a monodisperse state, there is an increase in the efficiency of covalent binding and enhanced luminescence ability between HSA and dye molecule (**Figure R6b, c**).

Figure R6. (a) Flow chart exploring the relationship between covalent binding efficiency and the number of dyes with monomer form. (b) NIR-II brightness after reaction of the HSA and CO-1080 in DMSO and CO-1080 in PBS. (c) SDS-PAGE gel electrophoresis after reaction of the HSA and CO-1080 in DMSO and CO-1080 in PBS.

3. The observed value of dye fluorescence lifetime is very short which is common for aggregates.

The increase in fluorescence intensity from 40k (noncovalent) to 200k(covalent) for CO-1080 with no significant change in fluorescence lifetime for equal amount of compound. Is it possible to add to non-covalent and covalent binding with HSA some amount of tween-80 or related reagent to confirm mechanism of disaggregation, provided in article?

Response: Thanks for providing this important suggestion. According to the reviewers' comment, we attempted to add certain amount of tween-80 to the non-covalent and covalent binding system with HSA to confirm the mechanism of disaggregation. The specific experimental details were shown in **Figure R7**. Results showed that the addition of tween-80 to the non-covalent and covalent binding system with HSA did not have a significant effect on the NIR-II brightness and non-covalent/covalent binding of the HSA@CO-1080 complex. (**Figure R8**). This suggests that the increase in fluorescence intensity of CO-1080 from 40k (non-covalent) to 200k (covalent) is primarily due to the effective torsional restriction of the protein on the dye molecule. In addition, the highly efficient and controllable one-to-one precise binding between

protein and dye successfully dissociated the clustered state of the dye, thus achieving a significant increase in brightness.

Figure R7. Flow chart for the mechanism exploration of disaggregation.

Figure R8. (a) NIR-II brightness and (b) gel electrophoresis of non-covalent and covalent binding between CO-1080 and HSA with or without addition of tween-80.

4. The value of measured singlet oxygen generation in solutions of protein without dye like blank standard samples and in nonpolar aprotic solvent like toluene can significantly improve the quality if manuscript.

Response: Thanks for providing this important feedback. According to the reviewers' comments, we have added the singlet oxygen production data of blank protein solutions without dyes and CO-1080 in nonpolar aprotic solvent (**Figure R9**).

Figure R9. Singlet oxygen (1O_2) generation of (a) blank protein (HSA) solutions without dye and (b) CO-1080 in toluene (DPBF as a probe).

Reviewers' Comments:

Reviewer #2:

Remarks to the Author:

All the concerns that I proposed have been properly addressed. The paper is recommended for publication in Nature Communications.

Reviewer #3:

Remarks to the Author:

No remarks

minor: l62 chromophobe, at l61 corrected
l966 and after - no of description for figures

Point by point replies to reviewers and revisions made.

Reviewer #3 (Remarks to the Author):

No remarks

**minor: l62 chromophobe, at l61 corrected
1966 and after - no of description for figures**

Response and revisions:

We thank the reviewers for endorsing our last round of revisions. We also highly appreciate the reviewer's suggestions for further strengthening our work. According to the reviewer' comments, we have rechecked the manuscript and made the corresponding error correction.